# MODIS Vegetation Continuous Fields tree cover needs calibrating in tropical savannas.

Rahayu Adzhar[1,2,3]*, Douglas I Kelley[1]*, Ning Dong[3,4], Charles George[1], Mireia Torello Raventos[5], Elmar Veenendaal[6], Ted R Feldpausch[7], Oliver L Phillips[8], Simon L Lewis[8,9], Bonaventure Sonké[10], Herman Taedoumg[11], Beatriz Schwantes Marimon[12], Tomas Domingues[13], Luzmila Arroyo[14], Gloria Djagbletey[15], Gustavo Saiz[16], France Gerard[1]

[1] U.K. Centre for Ecology & Hydrology, Wallingford, Oxfordshire, U.K.
[2] Department of Ecology, Evolution, and Environmental Biology, Miami University, Ohio, U.S.A
[3] Department of Life Sciences, Imperial College London, Berkshire, U.K.
[4] Department of Biological Sciences, Macquarie University, North Ryde, Australia
[5] School of Earth and Environmental Science, James Cook University, Cairns, Australia
[6] Plant Ecology and Nature Conservation, Wageningen University, Wageningen, The Netherlands
[7] College of Life and Environmental Sciences, University of Exeter, Exeter, U.K.
[8] School of Geography, University of Leeds, U.K.
[9] Department of Geography, University College London, London, U.K.
[10] Plant Systematics and Ecology Laboratory, Department of Biology, Higher Teachers' Training College, University of Yaoundé, Yaoundé, Cameroon
[11] Consultative Group on International Agricultural Research | CGIAR · Bioversity International, Yaoundé, Cameroon
[12] State University of Mato Grosso, Mato Grosso, Brazil
[13] Departamento de Biología (Ribeirão Preto), University of São Paulo, São Paulo, Brazil
[14] Universidad Autónoma Gabriel René Moreno, Santa Cruz, Bolivia
[15] Forest and Climate Change Division, Forestry Research Institute of Ghana, Ghana
[16] Facultad de Ciencias, Universidad Católica de la Santísima Concepción, Concepción, Chile

*Correspondence to:* adzharrb@miamioh.edu; doukel@ceh.ac.uk

**Abstract.** The Moderate Resolution Imaging Spectroradiometer vegetation continuous fields (MODIS VCF) Earth observation product is widely used to estimate forest cover changes, parameterise vegetation and Earth System models, and as a reference for validation or calibration where field data are limited. However, although limited independent validations of MODIS VCF have shown that MODIS VCF's accuracy decreases when estimating tree cover in sparsely-vegetated areas such as tropical savannas, no study has yet assessed the impact this may have on the VCF-based tree cover data used by many in their research. Using tropical forest and savanna inventory data collected by the TROpical Biomes in Transition (TROBIT) project, we produce a series of calibration scenarios that take into account (i) the spatial disparity between the in-situ plot size and the MODIS VCF pixel, and (ii) the trees' spatial distribution within in-situ plots. To identify if a disparity also exists in products trained using VCF, we used a similar approach to evaluate the finer-scale Landsat Tree Canopy Cover (TCC) product. For MODIS VCF, we then applied our calibrations to areas identified as forest or savanna in the International Geosphere-Biosphere Programme (IGBP) land cover mapping product. All IGBP classes identified as 'savanna' show substantial increases in cover after calibration, indicating that the most recent version of MODIS VCF consistently underestimates woody cover in tropical savannas. We also found that these biases are propagated in the finer-scale Landsat TCC. Our scenarios suggest that MODIS VCF accuracy can vary substantially, with tree cover underestimation ranging from 0 to 29 %. Models that use MODIS VCF as their benchmark could therefore be underestimating the carbon uptake in forest-savanna areas

and misrepresenting forest-savanna dynamics. Because of the limited in-situ plot number, our results are designed to be used as an indicator of where the product is potentially more or less reliable. Until more in-situ data are available to produce more accurate calibrations, we recommend caution when using uncalibrated MODIS VCF in tropical savannas.

## 1 Introduction

Tree cover values derived from Earth observation (EO) data form a fundamental part of ecological research. They are used to estimate forest cover change, biomass, and carbon stocks (Bastin et al., 2019; Giriraj et al., 2017; Saatchi et al., 2011; Song et al., 2014); help identify key areas for conservation efforts (Miles et al., 2006); and are used as a basis for climatic and vegetation modelling and model evaluation (Brovkin et al., 2013; Burton et al., 2019; Kelley et al., 2013). All this research, in turn, plays a vital role in informing local, regional, and global environmental policies (Harris et al., 2012). As such, an EO product's accuracy is important to consider, as any errors in the initial tree cover estimate can be further compounded in downstream work.

Only a handful of EO products provide global maps of percentage tree cover or forest and shrub cover distributions (Bartholomé and Belward, 2005; Bicheron et al., 2008), and fewer still provide information stretching over at least a decade (Friedl et al., 2002; Hansen et al., 2003, Sexton et al., 2013, DiMiceli, 2017). Of these, one of the products most widely used in ecological modelling is the Moderate Resolution Imaging Spectroradiometer Vegetation Continuous Fields (MODIS VCF) product (DiMiceli, 2017). MODIS VCF is a yearly product that provides percent tree cover globally at a spatial resolution of 250 m. The most recent iteration (Collection 6) is available for the years 2000 through to 2020. Its quantitative measure of woody cover is recorded annually and is described as a percentage of ground cover, making it particularly suited for use in evaluating dynamic global models (Lasslop et al., 2018; Rabin et al., 2017), as a proxy for in-situ data that are harder to collect (Kelley et al., 2019), and to help define parameters for calculating global tree restoration potential (Bastin et al., 2019). MODIS VCF is also used to train alternative products, such as the newer finer-scale Landsat Tree Canopy Cover (TCC) product (Sexton et al., 2013).

As the VCF product has progressed from Collection 1 to its current Collection 6, several validations using in-situ field data or higher-resolution remotely sensed data as a reference measurement have been carried out. These have been few and limited to sites within a biome (Montesano et al., 2009a), a region (Hansen et al., 2005; White et al., 2005), or within a country (Gao et al., 2014; Sexton et al., 2013). The MODIS VCF product evaluated was the most recent collection available at the time (i.e. Hansen et al., 2005 and White et al., 2005 for Collection 3; Montesano et al., 2009a for Collection 4; and Gao et al., 2015 and Sexton et al., 2013 for Collection 5). To our knowledge, no such independent validation experiment has yet been conducted on Collection 6, which produces tree cover estimates in the same manner as Collection 5 but with improvements made to the upstream inputs to enhance its accuracy (DiMiceli, 2017). Likewise, validation of the finer-scale TCC product has been limited to its penultimate version and to the taiga-tundra circumpolar region (Montesano et al., 2016).

The validations found that MODIS VCF may be less suitable for estimating tree cover in sparsely-vegetated areas. Huang & Siegert (2006) noted that MODIS VCF classified large areas of land as 'bare' where their land cover classification system identified it as sparsely-vegetated. Montesano et al. (2009) found that MODIS VCF data (Collection 4) overestimated cover in areas of low tree cover in taiga-tundra transition zones. Sexton et al. (2013) found that the Collection 5 product overestimated cover in areas of low cover (below 20 %) and underestimated in areas of higher tree cover, while Gao et al. (2015) found that MODIS VCF can only partially discriminate between tropical forest and non-forest, struggling in areas that have greater heterogeneity. Similarly to MODIS VCF (Montesano et al., 2009), Montesano et al., (2016) revealed an overestimation of the taiga-tundra low tree covers in the finer-scale Landsat TCC, suggesting that using VCF as training has propagated these overestimations into the higher resolution product. What is clear from the history of these validation and comparison experiments is that MODIS VCF has accuracy issues in areas with low woody vegetation cover, which has implications when its tree cover estimates are treated as accurately representative of real-world conditions. Failure to account for VCF's difficulty in estimating low woody covers can, therefore, lead to miscalibrated models and estimations that do not reflect real-world conditions. This, in turn, has knock-on effects on environmental policy-making, conservation efforts, and future ecological research, especially in areas with vegetation cover types that are most prone to error.

Tropical savannas have woody covers that fall within the range particularly affected by the reported MODIS VCF errors. A large proportion of these savannas can be found in tropical developing countries (Boval and Dixon, 2012), and are predicted to be home to half of the world's population by 2050 (State of the Tropics, 2020). Tropical savannas are therefore highly vulnerable to anthropogenic change. In the face of a growing population, land fragmentation, and changing climate, a savanna's ability to maintain robust ecosystem functions is directly linked to the amount of woody cover present (Sankaran et al., 2006). As a result, the ability to accurately monitor the state, dynamics, and woody cover trends of tropical savannas is a vital part of understanding how and why savannas are changing in the tropics (Harris et al., 2012; Miles et al., 2006), while also improving modelled climate projections and vegetation dynamics for this complex biome.

In this study, we evaluate MODIS VCF Collection 6 in tropical savannas and forest areas by comparing VCF's tree cover percentage to corresponding field data. Similarly, we evaluate Landsat TCC (version 4) to explore if, when VCF is used as training, VCF biases are propagated. We then, for MODIS VCF, characterise the observed bias in woody covers across both savanna and forest ecosystems and apply our calibration across the tropics to highlight the regions most likely affected by these inaccuracies. We finish by discussing the implications the uncovered biases may have on tropical vegetation and terrestrial biogeochemical modelling.

## 2 Methods

### 2.1 EO Products and Field data

We used the MODIS VCF Collection 6 product (250m spatial resolution, DiMiceli, 2017) with tree cover values averaged across the years 2006 through to 2009 to reflect the range of the field data collection period. MODIS VCF was downloaded using the modis r package (Hijmans, 2017) in R3.5.2 (R Core Team, 2018). We used the

2005 and 2010 30m Landsat TCC version 4 product (https://lcluc.umd.edu/metadata/global-30m-landsat-tree-canopy-version-4), and worked with the 2005 and 2010 average values. The product was downloaded manually

from https://e4ftl01.cr.usgs.gov/MEASURES/GFCC30TC.003/.

The in-situ field data were sourced from the 'TROpical Biomes InTransition' project (TROBIT) (www.geog.leeds.ac.uk/TROBIT, Torello-Raventos et al., 2013) and accessed via the Forestplots.net database (Lopez-Gonzalez et al., 2011; Lopez-Gonzalez et al., 2009). The data we used include the corner locations and

the Canopy Area Index (CAI) values for 17 forest and 31 savanna sites distributed across Australia, Brazil, Bolivia, Cameroon, and Ghana (Fig. 1 and Table A1, Fig. 2 in Torello-Raventos et al., 2013). The TROBIT field campaigns were carried out over a 3-year period, from 2006 to 2009, and the field plots used in this study are 1 hectare in size except for BFI-01 (0.5 ha), BFI-02 (0.5 ha), BFI-03 (0.5 ha), CTC-01 (0.93 ha), and VCR-01 (0.6 ha).

All the sites fall within the tropics, that is, within 23.5 degrees north and south of the equator, and were selected in regions where savannas and forests were in close proximity and within ecotones or 'zones of tension'. As such, the sites sampled show a large variation in physiognomy and edaphic and climatic conditions (Table S1, Veenendaal et al., 2015).

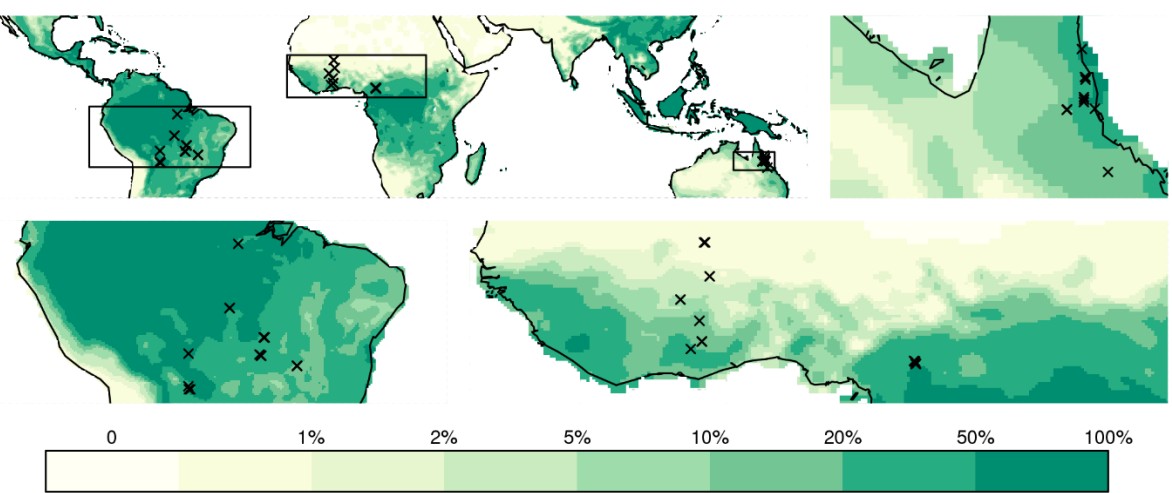

**Figure 1. Location of sampling sites in Africa, Australia, and South America from the TROBIT Project (based on Fig. 2, Torello-Raventos et al., 2013) shown on MODIS VCF (DiMiceli, 2017). Of the 63 field sites, only the 48 sites with available GPS coordinates were selected (https://www.forestplots.net).**

The classification of the TROBIT sites as either 'forest' or 'savanna' is based on the parameters described in Torello-Raventos et al. (2013) and Veenendaal et al. (2015). A 'savanna' is a natural land cover that is not a forest, bare ground, or a body of water. 'Forest' is defined as woody vegetation with an average tree height of or exceeding 6 m and a canopy area index (CAI) value of at least 0.3 for 'open forests' and 0.7 for 'forests'. In

addition, floristic differences (i.e. dominance of 'savanna' species) are used to differentiate forests from taller-growing savannas that have similar CAIs and tree heights (see Fig. 9, Torello-Raventos et al., 2013).

There is some ambiguity in how 'savannas' and 'grasslands' are defined. Some modelling-based research treat the two biomes as different (Whitley et al., 2017), while studies based on plant functional traits group them together (Solofondranohatra et al., 2018; White et al., 2000). As there is some concern that MODIS VCF will struggle to pick up woody cover in areas with really sparse vegetation, in this paper we decided to treat 'grasslands' as part of the savanna domain.

**2.2 Converting In-Situ Canopy Area Index to MODIS VCF / Landsat TCC percent tree cover**

CAI is defined as the sum of the projected areas of individual tree crowns divided by the ground area. In the TROBIT project (Torello-Raventos et al. (2013) and Veenendal et al. (2015)), plot-wide CAI is made up of the sum of the upper-stratum, mid-stratum, and subordinate-stratum crown areas. Membership to a stratum is determined by the tree's dbh (upper-stratum: dbh > 10 cm, mid-stratum: 2.5 cm < dbh < 10 cm, and subordinate-stratum: dbh < 2.5 cm, height > 1.5 m). About 50 trees per stratum per plot were measured to derive plot-specific allometric relations between stem diameter and crown area (supplement B of Torello Raventos et al., 2013). These were then applied to the whole plot to establish plot-level CAI. For the allometric relationships, tree crowns were treated as circles and the individual tree projected crown area was determined using the average of crown radii measured along the four cardinal points (i.e. from the centre of the stem to the distance furthest from the stem).

CAI values do not account for within-site tree canopy distribution patterns and the overlap between individual tree canopies. We account for this by converting each CAI value into a probability distribution function incorporating the following two extreme scenarios: "enforced overlap", where the location probability of individual canopies increases linearly from 0 to 1 across a site; and "unenforced overlap", where individual canopies follow a uniform random distribution pattern and canopy overlap is not purposefully introduced (Fig. 2). We repeated this 1000 times per CAI measurement to determine the probability distribution of expected CAI for each field plot.

Unlike CAI, which is the fraction of ground covered by tree crowns, the percent tree cover value from MODIS VCF (and so Landsat TCC) is defined as "the portion of the skylight orthogonal to the surface which is intercepted by trees" (Hansen et al. 2002). To make MODIS VCF and Landsat TCC comparable to tree cover derived from TROBIT plot CAIs, we divided these product values by 0.8 as suggested by Hansen et al. (2002). This is also the standard approach in most modelling studies using VCF (e.g Lasslop et al., 2020; Kelley et al., 2013; Burton et al., 2019). The 0.8 value can be thought of as a gap correction factor (GCF) that accounts for within-canopy gaps. Although the GCF has been shown to vary with vegetation type (Lloyd et al., 2008; 0.34 - 0.60) and crown cover (Tang et al., 2019: 0.70 - 0.96), we opted to use 0.8 as we found that it yielded more conservative results compared to a variable GCF. It also avoided introducing additional parameters into our analysis.

Next, to account for the difference in size between the MODIS VCF pixel (250 m x 250 m) and the smaller field plot size (100 m x 100 m), we calculated the possible percent tree cover an area the size of a TROBIT field plot could have, given the MODIS VCF percent tree cover for a MODIS-sized pixel. This was done for two extreme

scenarios: "enforced clumping," where all the tree cover for the given MODIS VCF value is forcibly 'clumped' on one side of the pixel, or "unenforced clumping," where 'clumping' is not enforced and tree cover is distributed randomly within the pixel (Fig. 3). The clumping scenarios introduce possible variations in percent cover due to the area and location mismatch between a TROBIT field plot and a MODIS pixel. A probability distribution was generated for each MODIS VCF pixel by calculating percent tree cover values for 1000 samples (100 m x 100 m) randomly placed within the 250 m x 250 m MODIS VCF pixel.

For Landsat TCC, where the Landsat TCC pixels (30m x30m) are smaller than the TROBIT field sites, we calculated a TCC percent tree cover to match the TROBIT field site size by summing the percent tree cover within the TCC pixel part found inside the TROBIT field site and then dividing the sum by the TROBIT site area.  As TROBIT site orientation was not recorded, we randomized the angle between the TROBIT site and TCC pixel grid for each of the1000 samples when generating the probability distribution. "Enforced clumping" was performed as per MODIS VCF (Fig 3), with the direction of clumping randomized.

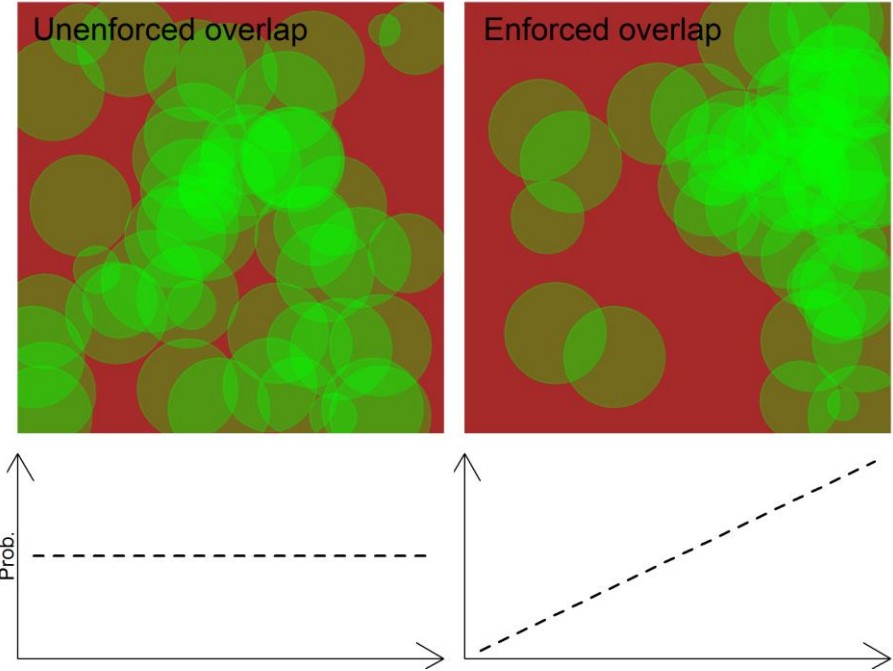

**Figure 2. Visual representation of the effects of enforcing overlap within a (100 m x 100 m) TROBIT site with a given Canopy Area Index  (CAI). Left: Overlap is not enforced, and individual crowns follow a uniform random distribution. Right: Overlap is enforced by linearly increasing the probability of a canopy being located more on one side of the site (i.e. here the right side of the site) than the other. This results in tree canopies 'overlapping' to a greater extent, which affects how accurately CAI represents actual canopy cover.**

### 2.3 Calculating Uncertainty Under Different Overlap-Clumping Scenarios

We compared both MODIS VCF and Landsat TCC with TROBIT under four different scenarios: 1) unenforced overlap and clumping; 2) enforce overlap and unenforced clumping; 3) unenforced overlap and enforced clumping; 4) enforced overlap and clumping. Comparisons were conducted by fitting the following logit function:

$$logit(Pixel) = C_0 + \Delta \times log(C^{\tau_1}/(1 - C^{\tau_2})) \quad \text{(Equation 1)}$$

Where $C_0, \Delta, \tau_1, \tau_2$ are optimised parameters and *Pixel* and *C* are the MODIS VCF / Landsat TCC pixel (post-conversion as described in section 2.2) and TROBIT site probability distributions, respectively. This is similar

to a standard linear regression of logit transformed data, accounting for maximum and minimum bounds of 0 - 100 % tree cover, with $\tau_1, \tau_2$ allowing for a non-symmetric transformation of tree cover. To account for the probability density of each point, we inferred the parameters in Equation 1 using a Total Least Squares Bayesian Inference technique using a Metropolis-Hastings Markov Chain Monte Carlo step. Priors were uninformed but physically bounded (i.e. $\Delta, \tau_1, \tau_2 > 0$) to assume an increasing relationship between MODIS VCF / Landsat

TCC and C. Equation 1 allowed us to assume normally distributed model errors, thus describing our conditional probability of observations for a given parameter combination by a normal distribution (Gelman et al., 2013). Each combination was run over 10 chains, with 1000 warm-up iterations and 10,000 sampling iterations. Optimisation was performed using the rstan2.19.2 (Stan Development Team, 2019) package in R3.5.2 (R Core Team, 2018). Our optimization accounts for potential errors in TROBIT cover, which includes those caused by

the allometric construction of the CAI, provided that the errors are unbiased and remain roughly consistent across sites (Gelman et al., 2013). As the TROBIT plots have relatively small total errors associated with the allometric relationships (Table B1, Torello-Raventos et al., 2013), systematic errors are unlikely to affect our results.

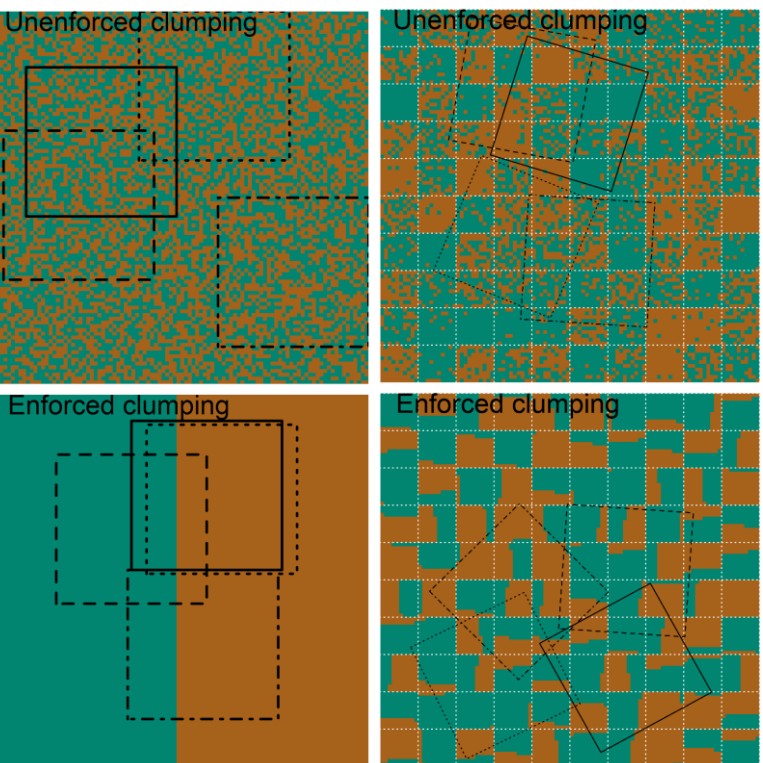

**Figure 3. Left: Example of the effects of unenforced and enforced clumping in a 250 m x 250 m MODIS VCF pixel with 50 % tree cover. Clumping all the cover (green) to one side of the pixel (left bottom) affects the average canopy cover value of a 100 m x 100 m-sized TROBIT site (black boxes). Right: Example of the effects of unenforced and enforced clumping on 30 m x 30 m Landsat TCC pixels with a mix of tree covers (green) and non-tree cover (brown). White dotted lines are TCC pixel boundaries. Clumping all the cover to one side of the pixel (right bottom) affects the**
**average canopy cover value of a 100 m x 100 m-sized TROBIT site (black boxes).**

**2.4 Mapping MODIS VCF Uncertainty Across The Tropics**

We evaluated the impact of the MODIS VCF biases inferred from these regression equations across the tropics
       by inverting our calculation of MODIS VCF bias (Fig. A1) as follows: first, the inverse (i.e. solving for C) of
       equation 1 was applied to MODIS VCF values after conversion to a 100 m x 100 m pixel size grid (matching the
       field site area); then this calibrated value was translated back to the original 250 m x 250 m VCF pixel size. As
       the inverse of Equation 1 has no analytical solution, we found the rounded percent value of C that minimises the

absolute difference between the left- and right-hand side of the equation. For computational feasibility, we
       constructed maps of the tropics with calibrated MODIS VCF values (Fig. A2) by randomly sampling 5 iterations
       from each of our 10 optimisation chains (50 in total) and masking out pixels with cover types not considered as
       'forest' or 'savanna' in the 500 m MODIS Land Cover Type (MCD12Q1 - collection 6) (Sulla-Menashe and
       Friedl, 2018).


       We then used the MCD12Q1 product to identify the areas of 'forest' and 'savanna' across the tropics in the
       MODIS VCF product. MCD12Q1 is widely used by the global land surface modelling community (e.g. Sellar et
       al., 2019; Wiltshire et al., 2020) and describes land cover in terms of 17 global land cover classes as per the
       International Geosphere-Biosphere Programme (IGBP, Table 3 in Sulla-Menashe and Friedl, 2018). The product

is based on the same spectroradiometer (MODIS) and temporal resolution as the VCF product. Referring to the
       definition of 'savanna' of Veenendaal et al. (2015), the following land cover classes were chosen to represent
       'savanna': Closed Shrubland, Open Shrubland, Woody Savanna, Savanna, and Grassland, while 'forest'
       encompasses: Evergreen Needleleaf Forests, Evergreen Broadleaf Forests, Deciduous Needleleaf Forests,
       Deciduous Broadleaf Forests, and Mixed Forests. We subset MCD12Q1 to the tropical zone between +/- 30°

North and took the median class for the 2006 to 2009 period, matching the field data collection period.

       For a more detailed land-cover-specific evaluation, we extracted the calibrated 250 m MODIS VCF pixel values
       for each corresponding 500 m MCD12Q1 pixel to construct land-cover-specific MODIS VCF tree cover
       frequency distributions (Fig. A3). Our tree cover calibration by cover type for the four clumping/overlap

regression combinations was then calculated by multiplying each cover type MODIS VCF frequency
       distribution (Fig. A3) with curves representing the median, 5 %, and 95 % confidence lines of the calibration
       equation ensembles.

       **3 Results**

**3.1 Comparing MODIS VCF and Landsat TCC to tree cover from TROBIT field sites**

       MODIS VCF underestimates tree cover within the 19 % to 81 % range across all four combinations of enforced-
       unenforced overlap and clumping (black line, Fig. 4). Below 12 %, MODIS VCF tree cover values do not
       significantly disagree with TROBIT field data, and may instead be overestimating tree cover (50 % confidence,
       dashed line, Fig. 4). A similar pattern is seen when tree cover exceeds 84 %: MODIS VCF does not differ

significantly from TROBIT when there is enforced overlap (i.e. when tree canopies are clustering towards one

side increasing the degree of canopy overlap - Fig. 4 right), but may underestimate tree cover when overlap is not enforced (i.e. tree canopies are spaced randomly within the site - Fig. 4 left).

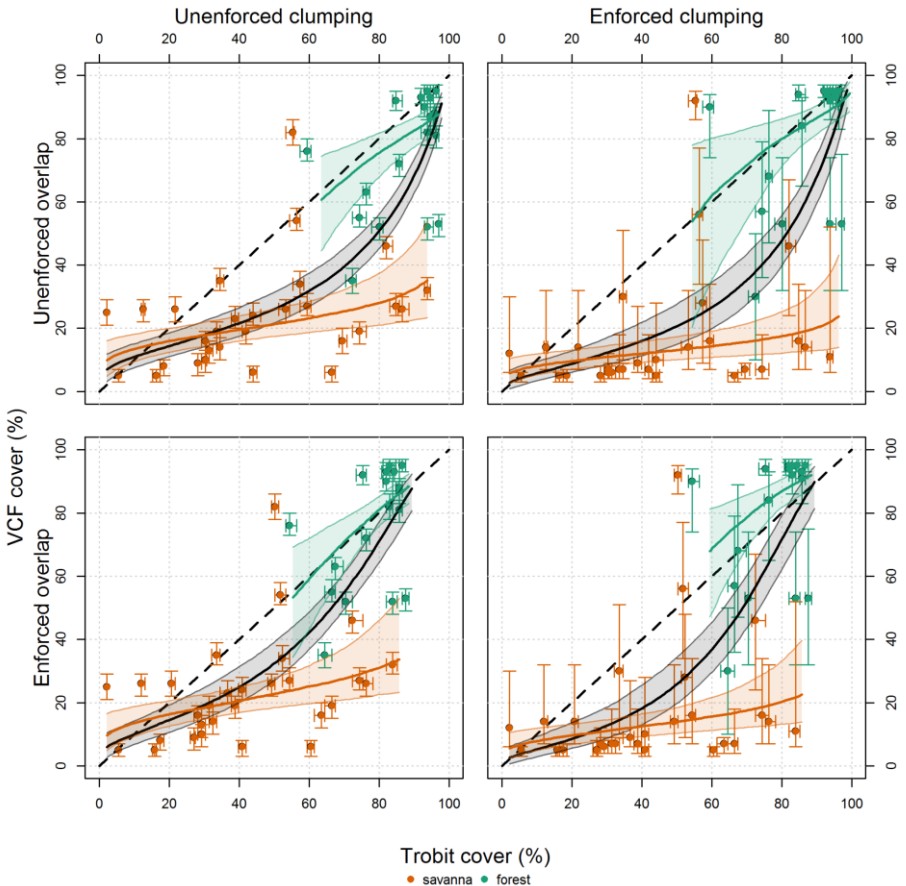

**Figure 4. MODIS VCF percent tree cover versus percent tree cover from TROBIT field data, taking into account uncertainties associated with tree cover spatial distributions within a MODIS pixel and/or field site. The 4 combinations are: (1) no overlap and no clumping, where tree canopies are randomly distributed within both pixel and site; (2) no overlap and maximum clumping, where tree canopies are clustered in one area of the pixel, and randomly distributed throughout the field site; (3) with overlap and no clumping, where tree canopies are randomly distributed within the pixel, but overlap substantially within the field site; and (4) with overlap and maximum clumping, where tree canopies are clustered to one side within a pixel, and overlap substantially within the site. The bolded dashed line in black shows the 1:1 relationship. The solid lines represent the median of the respective regressions (green for forest; orange for savanna; black for forest and savanna combined), and the thin lines represent the 5 and 95 % confidence interval of their respective regression lines. The vertical error bars represent uncertainty introduced by clumping; the horizontal error bars represent the uncertainty introduced by overlap.**

There is a clear difference in how accurately MODIS VCF estimates tree cover in forested areas (in green, Fig. 4) as opposed to areas identified as savannas (in orange, Fig. 4). In savanna sites, MODIS VCF significantly and consistently underestimates tree cover regardless of the amount of overlap and clumping. Significant underestimation (at 95 % confidence) occurs when *in-situ* tree cover exceeds 19-21% (without enforced clumping) or 11-12% (with enforced clumping). In forest sites, MODIS VCF does not show the same pattern of systematic underestimation. Divergence does occur at high covers, depending on the enforcement of overlap or clumping. MODIS VCF underestimates tree cover where tree cover exceeds 84 % (at the 95 % confidence interval) when neither overlap nor clumping is enforced, and overestimates where tree cover exceeds 78 % (at 5 % confidence interval) when both overlap and clumping are enforced.

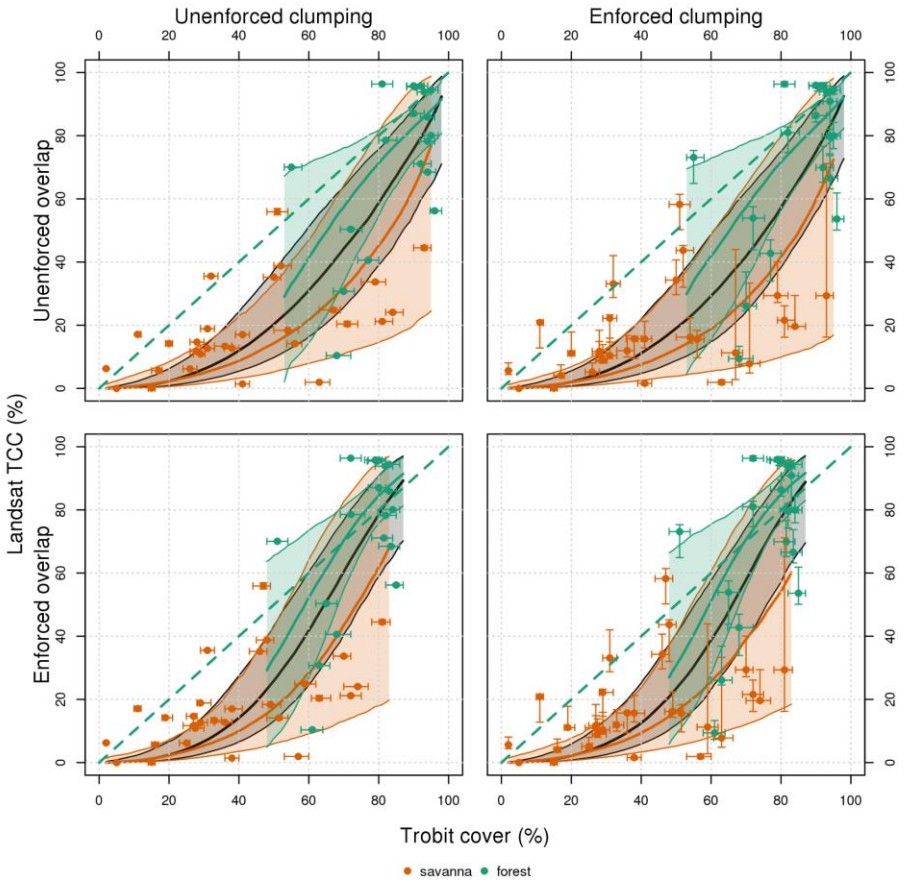

**Figure 5. Landsat TCC percent tree cover versus percent tree cover from TROBIT field data, taking into account uncertainties associated with tree cover spatial distributions within a TCC pixel and/or field site. The 4 combinations are: (1) no overlap and no clumping, where tree canopies are randomly distributed within both pixels and site; (2) no overlap and maximum clumping, where tree canopies are clustered in one area of the pixels, and randomly distributed throughout the field site; (3) with overlap and no clumping, where tree canopies are randomly distributed within the pixels, but overlap substantially within the field site; and (4) with overlap and maximum clumping, where tree canopies are clustered to one side within the pixels, and overlap substantially within the site. The bolded dashed line in black shows the 1:1 relationship. The solid lines represent the median of the respective regressions (green for forest; orange for savanna; black for forest and savanna combined), and the thin lines represent the 5 and 95 % confidence interval of their respective regression lines. The vertical error bars represent uncertainty introduced by clumping; the horizontal error bars represent the uncertainty introduced by overlap.**

Similar patterns can be observed with Landsat TCC (black line, Fig. 5). There is a significant underestimation of tree cover in the lower cover ranges up to 59% when there is enforced overlap, and up to 82% when overlap is not enforced. In savanna sites (orange line, Fig. 5) the underestimation (at 95% confidence) is significant and consistent for covers below 75 - 80 % (without enforced overlap) or below 52 - 60% (with enforced overlap). In forest sites (green line, Fig. 5) there is no systematic difference.

## 3.2 Global estimates of post-calibration change in tropical tree cover

We assessed the impact of VCF's underestimation of intermediate tree covers across the tropics (IGBP forest and savanna land cover classes), using a calibration based on the combined forest and savanna sites (black curve, Fig. 4) instead of using the savanna-only sites for a savanna-specific calibration (orange curve, Fig. 4). This is because there were few TROBIT sites representing savanna with MODIS VCF tree cover values exceeding 40 %, and global land cover maps disagree on the distribution of savannas within the forest-savanna ecotone (Herold et al., 2008).

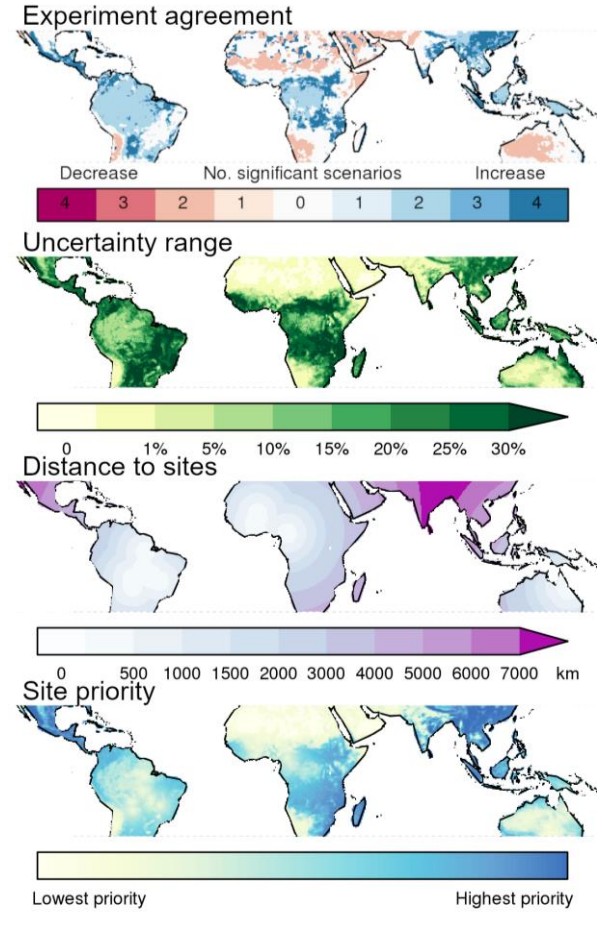

**Figure 6. (Top) the post-calibration change in tree cover that is statistically significant (95% interval) in the same direction (positive or negative calibration leading to an increase or decrease in tree cover, respectively) across all four scenarios. (upper middle) Uncertainty range of the post-calibration change in tree cover, calculated as the 90th percentile (maximum of the four scenarios in Fig A2) minus 10th percentile (minimum of the four scenarios in Fig. A2). (lower middle) Geographic distance to the closest TROBIT site sampled. (Bottom) Regions coloured to denote priority for field surveying to constrain map uncertainty (based on multiplying the uncertainty range of each pixel with the pixel's geographical distance to the closest TROBIT site sampled).**

The distribution of tree cover change after calibrating against field data are similar across the four scenarios (Fig. A2), and the regions where all four scenarios agree on the direction of change (positive and negative) are substantial (Fig. 6). However, there are some differences caused by the uncertainty introduced by different extents of overlap and clumping. While we see a significant increase in tree cover across all clumping-overlap combinations in many regions of tropical savannas and grasslands (Pennington et al., 2018), such as in the forest-savanna mosaics that surround Congolian rainforests, we do not see the same pattern in the Cerrado of Brazil. This is likely because the African forest-savanna regions fall within the range of MODIS VCF values that consistently undergo a positive calibration (~30 - 50 %, see Fig. A1), while the Cerrado of Brazil does not.

Using field plots over a limited geographic extent creates uncertainty that may be unaccounted for in our analysis when our calibration is broadly applied across the highly variable tropical forest-savanna ecotone. By multiplying the uncertainty range of our calibrations with the geographical distance to the closest sampled TROBIT site, we identified priority regions for further field surveying (Fig. 6 bottom). We found Southeast Asia, Central America, and Mexico are areas where additional in-situ observations would greatly help improve

confidence. Field data from the northwestern region of South America, the southeast of the African continent, and Madagascar would also help.

As our calibrations were based on a limited number of sites in a limited number of regions, it is important to note that the maps shown in Figures 6 and A2 are not definitive. For instance, we found a significant tree cover decrease in the Sahel post-calibration in multiple scenarios, which runs counter to the results of Brandt et al.

(2020) who found that tree cover was underestimated in this region. This disparity may be due to our lack of field sites in these more arid regions, further highlighting the importance of more in-situ data for more accurate and precise calibration. Therefore, our calibrations are most useful in identifying areas where MODIS VCF estimates may be more or less reliable.

### 3.3 Post-calibration change in tree cover within different vegetation classes in tropical ecosystems

When looking at our calibration in more detail, we see that MODIS VCF significantly underestimates tree cover in all the IGBP land cover classes that we considered, regardless of overlap or clumping (95 % confidence interval)(Fig. 7). The most substantial and significant underestimation is in the classes 'woody savannas' and 'savannas'. The underestimation is the largest in woody savannas, except when clumping and overlap are enforced (in purple, Fig. 7). This is because the peak in the tree cover frequency distribution for savannas aligns

with where the calibration for maximum overlap and clumping is the largest (i.e. at about 20 % tree cover, see Fig. A3), while the peak in cover distribution for woody savannas (26 - 67 %, Fig. A3) aligns with the cover range that undergoes the greatest change ( Fig. 7) in the other clumping and overlap scenarios.

'Open shrublands' only show a small underestimation of tree cover, despite its woody cover definition (10 - 60 %) matching the range where MODIS VCF most underestimates tree cover (26 - 67 % cover). The discrepancy

may be because the majority of the 'open shrublands' class commission error is with the 'grasslands' class (see Table S6 in Sulla-Menashe et al., 2019). The MODIS VCF tree cover in areas classified as 'open shrublands' is therefore likely to be lower than the IGBP definition would suggest (see Fig. A3), resulting in calibrations that are more conservative.

We found significant increases in tree cover for 'forests' in every calibration scenario, though net change is not

significant (95 % confidence) when overlap is enforced. This can be explained by the presence of both negative and positive calibrations in the higher ranges of tree cover when overlap is enforced. Similarly, the net change is insignificant across all clumping and overlap scenarios for the IGBP classes matching the lower ranges of tree cover (grassland, close shrubland and open shrubland).

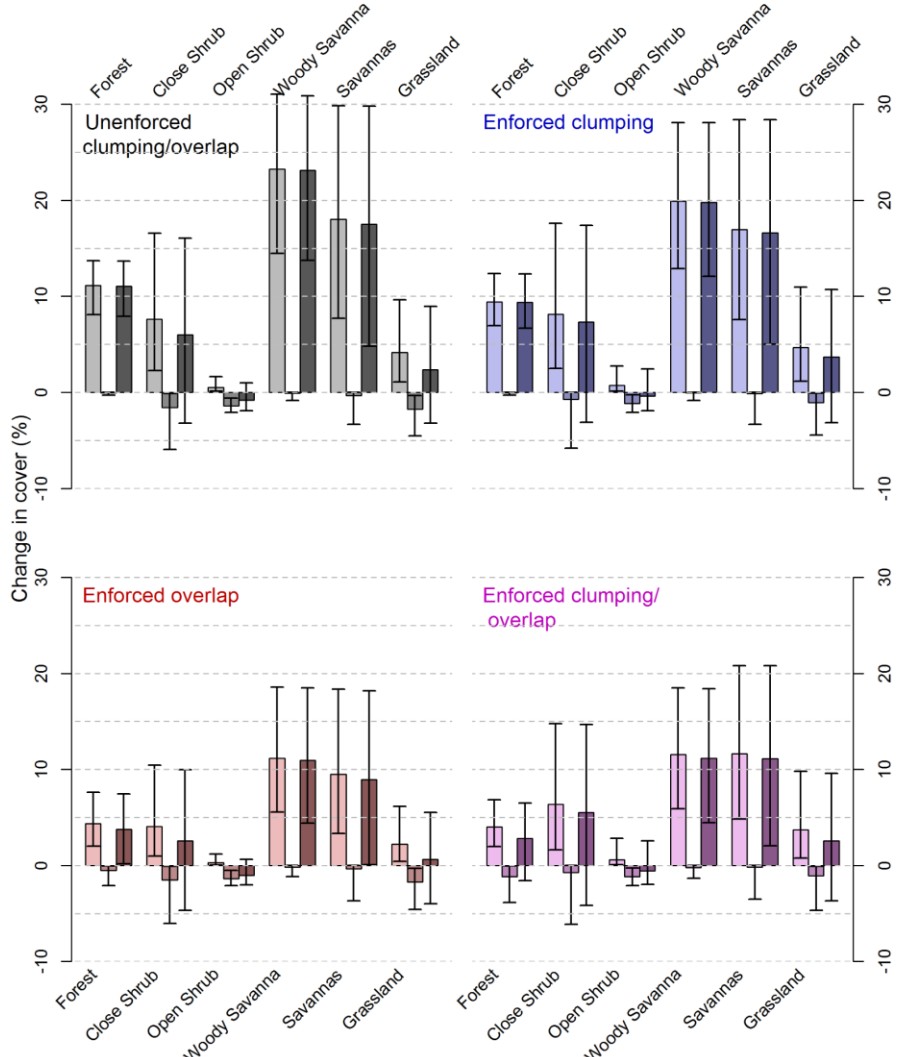

**Figure 7. Percent change in tree cover post-calibration (clockwise: no enforced clumping or overlap (black); enforced clumping and no enforced overlap (blue); no enforced clumping and enforced overlap (red); enforced clumping and overlap (pink) in the 'forest' supercategory and the 5 savanna classes. Palest tone indicates positive change, mid-tone indicates negative change, and the darkest tone indicates net change. Error bars denote the 5-95 % confidence interval; if the error bar extends past the x-axis, the post-calibration change is not considered significant.**

## 4 Discussion

While MODIS VCF is a powerful and accessible tool to map tree cover, our field data-based calibrations indicate that the latest MODIS VCF collection 6 is missing a lot of woody cover, even when uncertainty

introduced by site canopy overlap and clumping within the MODIS VCF pixel are accounted for. The Landsat TCC product, which may be viewed as an alternative with a higher spatial resolution, behaves in a similar manner. Our map (Fig. 6, top) highlights that this potential underestimation of woody cover is mainly occurring in tropical savannas. Moreover, the highest underestimation in the savanna classes occurs when there is no enforced overlap (Fig. A2) (i.e. when there is a uniform random distribution of trees) which is the scenario that

most likely reflects the TROBIT savanna plots. This is evidenced by work done by Veenendaal et al. (2015), where TROBIT plots were tested for complete spatial randomness and only minor indications of overlap were

found. Woody savannas, as an example, may have their tree cover underestimated by up to 32 % (95 % confidence) when neither clumping nor overlap is enforced (grey tones, Fig. 7). If our results are representative of the tropics, then overall, MODIS VCF may be underestimating tropical tree cover by between 7 - 29 % for unenforced clumping and overlap or 0 - 21 % for when either clumping or overlap are enforced (5 - 95 % confidence).

An overestimation at the lower end of the cover (< 20 %) (Hansen et al., 2002; Sexton et al., 2013) and underestimation in the lower to middle range of cover (20 % - 60 %) have been identified in validations of previous MODIS VCF collections (Gross et al., 2018; Yang and Crews, 2019) and Landsat TCC version (Montesano et al. 2016). According to its definition, MODIS VCF only maps trees that are 5 m or taller (Hansen et al. 2003), while the TROBIT CAI includes all trees with a minimum dbh of 2.5 cm, as well as trees with a height exceeding 1.5 m when dbh < 2.5 cm. This could explain our observed underestimation in the lower tree cover ranges for both MODIS VCF and Landsat TCC. In fact Montesano et al. (2016) showed an improved match between Landsat TCC and their lidar-derived tree cover reference data when reducing the height threshold from 5 m to 2 m. However, because of how our field reference CAI is derived, we were not able to conclusively link the 5 m threshold to our observed underestimation.

On the other hand, when looking at the relationship between TROBIT's upper stratum canopy height and the difference between TROBIT and VCF we would have expected an increasing underestimation in the lower height ranges. Instead we found a low $R^2$ and a mixture of under and overestimations in heights between 0 and 10 m (Fig. A4). This suggests that the inclusion of trees below 5 m height in the TROBIT inventory does not fully explain the observed underestimation. However, as the relationship between upper canopy heights and the subordinate strata composition (and canopy cover thereof) varies widely depending on factors including ecosystem type and altitude (Rutten et al., 2015), more research needs to be done with in-situ height data.

We also found discrepancies between the tree cover values derived from MODIS VCF and the corresponding class definition of the MCD12Q1 product (Fig. A3) which again suggests that the 5 m height threshold may not always apply in MODIS VCF. For example, MODIS VCF recorded tree cover in the 'open shrublands' and 'closed shrublands' classes of the MCD12Q1 product (Fig. A3), even though the height range for these classes is 1 - 2 m. For the 'savannas' class, MODIS VCF yields a percent tree cover range that matches closely with the 'savannas' class definition (between 10 % and 30 %), despite the differing tree thresholds for MODIS VCF and IGBP (5 m minimum for MODIS VCF, and 2 m minimum for IGBP). These discrepancies suggest one of the following three things: 'open/closed shrublands' and 'savannas' contain trees taller than 5 m; MODIS VCF is distinguishing trees below the 5 m threshold; or, some combination of both.

Another explanation for the discrepancy between the IGBP class definitions and those estimated through MODIS VCF could be the between-class omission and commission errors (Fig. 5, and Table S6 in Sulla-Menashe et al., 2019). For example, the accuracy for 'closed shrublands' is particularly low. It is mainly confused with 'open shrublands', 'woody savannas' and 'savannas'. The majority of the 'open shrublands' class commission error is with the 'grasslands' class and there is confusion to a lesser extent between 'open shrubland', 'woody savannas' and 'savannas'. Also, the 'cropland/natural vegetation mosaics' class is often mapped as 'closed shrubland', 'woody savannas', 'savannas' or 'grasslands'.

More work needs to be done to evaluate how effective both MODIS VCF, Landsat TCC and MCD12Q1 are at implementing the height thresholds in their respective 'tree' definitions, as this may have implications when MODIS VCF, Landsat TCC and MCD12Q1 are used for global model calibration or validation.

Overall, our results suggest that the biases found in the previous collections may have persisted in collection 6, despite reported improvement in accuracy (DiMiceli et al., 2017). This indicates that the biases introduced by binning the training data (Gerard et al. 2017) and using a CART (Classification and Regression Tree) model (Hanan et al., 2013) are inherent and still present within this version of MODIS VCF.  Similar results for MODIS VCF and Landsat TCC also suggest that by training TCC with VCF tree cover these biases have been propagated into the finer-scale product.  There is a risk of bias propagation when MODIS VCF (or related products) is used as a single source for benchmarking models (e.g.  Brandt et al., 2017; Lasslop et al., 2020; Burton et al., 2019; Kelley et al., 2019, 2021). To avoid this, studies should try use multiple data sources (e.g. for woody cover possibly the GEDI lidar based canopy cover product, Tang et al 2019) whether for model calibration/validation (e.g. Sellar et al., 2019 and Wiltshire et al., 2020, Burton et al., 2021)  or hypothesis testing (e.g. Taylor et al., 2012).

We suggest that, while MODIS VCF and landsat TCC give a good overview of tree cover on a global scale, both should be used cautiously in savanna regions. Special care should be taken in savannas, a biome that has long been noted as being challenging for EO products to characterise, as solitary trees in the landscape tend to be missed by global tree cover products (Jung et al., 2006, Brandt et al., 2020). The poor performance of MODIS VCF and Landsat TCC in savannas in particular (Gaughan et al., 2013; Gross et al., 2018; Kumar et al., 2019) emphasises the importance of continuous independent validation and re-calibration of these products. The ecosystem functions of savannas can vary drastically with just a slight difference in tree cover (Gaughan et al., 2013) and even slight errors may create issues in how we interpret the state and dynamics of the biome, which in turn affects how the land is managed.

Work on forest restoration potential would also be impacted. Bastin et al. (2019), for example, used MODIS VCF to estimate tree cover in agricultural land. As this tree cover is likely to have been underestimated substantially, the derived available land space for replanting may be less than projected, with the restoration potential overestimated. However, our results also indicate an underestimated tree cover in woodier savannas and forests. Accounting for this, the restoration potential could actually be greater than anticipated, because the carrying capacity of a unit of land could be greater than previously thought. Calibration could also result in a more uniform cover distribution across regions, producing a more gradual transition between low-cover savannas and high-cover forests. This could have implications for work that, for example, uses MODIS VCF to study forest-savanna dynamics and bi-stability (Lasslop et al., 2018; Wuyts et al., 2017; Xu et al., 2016).

To ensure the appropriate use of both products, we suggest that where field data are available, the products should be calibrated for use in the target region. However, calibrating on a large scale using field data as a reference presents several challenges. Firstly, different in-situ measurement techniques tend to measure different types of tree cover (e.g. Fiala et al., 2006; Korhonen et al., 2006; Rautiainen et al., 2005) and each will require a specific conversion method to enable direct comparison with  MODIS VCF or Landsat TCC. For example,

Montesano et al. (2016) 's comparison did not acknowledge VCF's and thus TCC's 'within canopy gaps,' which may explain their observed underestimation in covers above 80%. In our case, to account for gaps between tree crowns, we applied the 0.8 'gap correction factor' to the CAI. However, the GCF and resulting tree cover could vary widely on a plot-by-plot basis (Lloyd et al., 2008). With further in-situ data that describe tropical vegetation type-specific GCF variation, we may be able to incorporate site-specific GCFs into our analysis.

There is also the uncertainty associated with the field data collection. In our case, the site-specific CAI standard errors (supplement B in Torello Raventos et al., 2013) are small and show no systematic bias, and are therefore not expected to significantly change our results. However, our results in Figure 6 have been extrapolated from a limited number of field sites, with somewhat limited distribution across the tropics. While our uncertainty map gives a good idea of areas of concern, for a more robust description of MODIS VCF's accuracy, we would need substantially more widely distributed in-situ sites to represent the tropics. Field data collection remains costly and labour intensive and adopting protocols that ensure standardisation and data sharing via information platforms would facilitate large-scale validation exercises. In the case of validating woody cover products, including direct tree cover measurements such as crown area would also help.

However, the reality for much of pre-existing data is that they are not standardised, and field data (whether standardised or not) are not designed to be directly comparable to remote sensing-derived variables. Here, our technique offers a potential solution to this. By accounting for uncertainties that arise from differences between the available in-situ data and the remotely sensed variable we are able to use in-situ data, which may initially seem unsuitable, to carry out a more conservative evaluation.

Finally, factors such as cloud cover, landscape heterogeneity, phenology, vegetation type, and soil type affect the accuracy of remotely-sensed products like MODIS VCF and Landsat TCC (Hansen et al., 2003; Huete et al., 1997; Smith et al., 2002). Data characterising these at the plot level would help identify potential confounding factors affecting performance, and so help further constrain uncertainties.

Alternatively, comparing MODIS VCF to other land cover maps or higher-resolution remotely sensed data are recommended (Gross et al., 2018; Lary and Lait, 2006, Montesano et al. 2016), though without a large-scale effort to re-calibrate MODIS VCF and products trained using VCF like Landsat TCC, the question of how appropriate MODIS VCF is for use in both forests and savannas in the tropics will remain. By highlighting the extent to which MODIS VCF struggles to estimate tree cover in tropical forests and savannas, we hope to inform the future use of this product and improve its useability.

**5 Conclusion**

We found that MODIS VCF significantly underestimates tree cover in tropical forests and savannas, even when within-site and site-pixel variation are accounted for in calibration. We also found that using MODIS VCF for training likely propagates these biases, even in the finer-scale Landsat TCC. As MODIS VCF is a product that is commonly used in a wide variety of ecological research including vegetation modelling, estimating restoration potential, and identifying forest-savanna bimodality, we stress that more independent work on validating and re-calibrating is required before its tree cover estimates can be relied upon in the tropics.

 **Appendix**

| Site Name | Country | Latitude | Longitude | MODIS VCF Tree Cover (%) | Canopy Area Index | Average Upper Stratum Height (m) | Cover Type | TROBIT Site Description |
|---|---|---|---|---|---|---|---|---|
| ALC-01 | Brazil | -2.53 | -54.91 | 12.5 | 0.32 | 6.56 | Savanna | Savanna woodland |
| ALF-01 | Brazil | -9.6 | -55.94 | 77 | 2.31 | 37.02 | Forest | Tall forest |
| ALF-02 | Brazil | -9.58 | -55.92 | 76 | 2.65 | 41.32 | Forest | Tall forest |
| ASU-01 | Ghana | 7.14 | -2.45 | 41.33 | 2.54 | 45.27 | Forest | Tall forest |
| BBI-01 | Burkina Faso | 12.73 | -1.17 | 1.33 | 0.52 | 12.53 | Savanna | Savanna woodland |
| BBI-02 | Burkina Faso | 12.73 | -1.16 | 1.5 | 0.99 | 13.6 | Savanna | Savanna woodland |
| BDA-01 | Burkina Faso | 10.94 | -3.15 | 6.17 | 0.3 | 14.53 | Savanna | Shrub-rich savanna woodland |
| BDA-02 | Burkina Faso | 10.94 | -3.15 | 4.5 | 0.18 | 14.47 | Savanna | Shrub-rich savanna woodland |
| BFI-01 | Ghana | 7.71 | -1.69 | 15 | 1.22 | 29.67 | Savanna | Tall closed woodland |
| BFI-02 | Ghana | 7.71 | -1.69 | 12.83 | 1.08 | 28.2 | Savanna | Tall savanna woodland |
| BFI-03 | Ghana | 7.71 | -1.7 | 25.83 | 2.54 | 45.07 | Savanna | Tall savanna woodland |
| CTC-01 | Australia | -16.1 | 145.45 | 72.67 | 2.35 | 40.37 | Forest | Tall forest |
| DCR-01 | Australia | -17.02 | 145.58 | 21.67 | 1.67 | 27.19 | Savanna | Tall savanna woodland |
| DCR-02 | Australia | -17.03 | 145.6 | 65.67 | 0.71 | 22.51 | Savanna | Tall savanna woodland |
| EKP-01 | Australia | -18.07 | 145.99 | 43.5 | 0.74 | 28.13 | Savanna | Tall savanna woodland |
| FLO-01 | Brazil | -12.81 | -51.85 | 65.67 | 2.4 | 28.21 | Forest | Forest |
| FMS-01 | Australia | -18.09 | 144.84 | 7.67 | 0.32 | 20.03 | Savanna | Shrub-rich savanna woodland |
| FMS-02 | Australia | -18.11 | 144.82 | 44.17 | 1.21 | 16.69 | Forest | Stunted shrub-rich forest |
| HOM-01 | Mali | 15.34 | -1.47 | 0.5 | 0.05 | 3.87 | Savanna | Savanna grassland |
| HOM-02 | Mali | 15.33 | -1.55 | 0.83 | 0.16 | 6.13 | Savanna | Savanna grassland |
| IBG-01 | Brazil | -15.95 | -47.87 | 20.83 | 0.22 | 7.48 | Savanna | Scrub savanna |
| IBG-02 | Brazil | -15.95 | -47.87 | 20 | 0.02 | 6.29 | Savanna | Scrub savanna |
| IBG-03 | Brazil | -15.93 | -47.87 | 20.5 | 0.12 | 8.01 | Savanna | Scrub savanna |
| IBG-04 | Brazil | -15.94 | -47.86 | 27.17 | 0.77 | 12.65 | Savanna | Savanna woodland |
| KBL-01 | Australia | -17.77 | 145.54 | 75 | 1.69 | 39.5 | Forest | Tall forest |
| KBL-02 | Australia | -17.85 | 145.53 | 61.17 | 0.81 | 29.2 | Savanna | Tall savanna woodland |
| KBL-03 | Australia | -17.69 | 145.53 | 79.5 | 3 | 36.62 | Forest | Tall forest |
| KCR-01 | Australia | -17.11 | 145.6 | 78.83 | 2.44 | 42.37 | Forest | Tall forest |
| LFB-03 | Bolivia | -14.6 | -60.85 | 28.17 | 0.39 | 9.93 | Savanna | Shrub-rich savanna woodland |
| MDJ-01 | Cameroon | 6.17 | 12.83 | 42 | 3.24 | 45 | Forest | Tall forest |
| MDJ-02 | Cameroon | 6.16 | 12.82 | 18.67 | 0.44 | 16.13 | Savanna | Long-grass savanna |

| | | | | | | | | |
|---|---|---|---|---|---|---|---|---|
| MDJ-03 | Cameroon | 5.98 | 12.87 | 64.67 | 2.97 | 36.53 | Forest | Stunted shrub-rich forest |
| MDJ-04 | Cameroon | 6 | 12.87 | 15 | 0.37 | 18.93 | Savanna | Long-grass savanna |
| MDJ-05 | Cameroon | 5.98 | 12.87 | 70.33 | 2.85 | 21.27 | Forest | Stunted shrub-rich forest |
| MDJ-06 | Cameroon | 6 | 12.89 | 20.5 | 0.68 | 15.27 | Savanna | Long-grass savanna |
| MDJ-07 | Cameroon | 6.01 | 12.89 | 57.33 | 1.75 | 42.67 | Forest | Tall forest |
| MDJ-08 | Cameroon | 6.21 | 12.75 | 15 | 0.48 | 18 | Savanna | Long-grass savanna |
| MLE-01 | Ghana | 9.3 | -1.86 | 10 | 0.34 | 14.67 | Savanna | Savanna woodland |
| NXV-02 | Brazil | -14.7 | -52.35 | 20.83 | 1.82 | 15.76 | Savanna | Tall closed woodland |
| RSC-01 | Australia | -20.16 | 146.54 | 28 | 1.15 | 13.14 | Forest | Stunted forest |
| SMT-01 | Brazil | -12.82 | -51.77 | 36.67 | 1.55 | 14.37 | Savanna | Savanna woodland |
| SMT-02 | Brazil | -12.82 | -51.77 | 41.5 | 1.44 | 14.64 | Savanna | Savanna woodland |
| SMT-03 | Brazil | -12.83 | -51.77 | 19.33 | 0.53 | 11.19 | Savanna | Savanna woodland |
| TUC-01 | Bolivia | -18.52 | -60.81 | 50.33 | 1.29 | 14.9 | Forest | Stunted forest |
| TUC-02 | Bolivia | -18.53 | -60.63 | 21.67 | 0.81 | 12.05 | Savanna | Shrub-rich woodland |
| TUC-03 | Bolivia | -18.19 | -60.86 | 10.83 | 0.37 | 14.11 | Savanna | Savanna woodland |
| VCR-01 | Brazil | -14.83 | -52.16 | 69.5 | 2.81 | 28.94 | Forest | Tall forest |
| VCR-02 | Brazil | -14.83 | -52.17 | 69.67 | 2.74 | 30.93 | Forest | Forest |

**Table A1. Site names, locations, Canopy Area Index values, MODIS VCF percent tree cover values, cover type, and TROBIT site descriptions of the 48 TROBIT Project plots used in this study. TROBIT site descriptions are based on Table S1 of Veenendaal et al., 2015.**


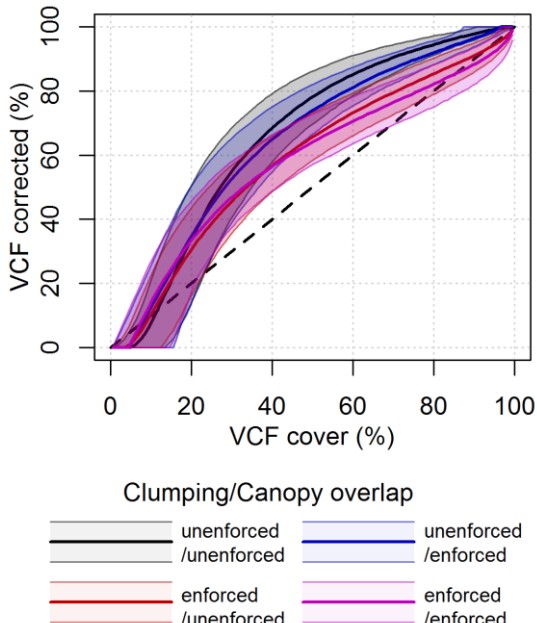

**Figure A1. The calibration curves developed for MODIS VCF based on 4 pixel-site mismatch scenarios (no clumping and no overlap; enforced clumping no overlap; no clumping enforced overlap; and enforced clumping and enforced overlap). The dashed line signifies the 'ideal' 1:1 relationship wherein calibrated MODIS VCF is unchanged from the original MODIS VCF values. The shaded regions represent 5 to 95 % confidence intervals for the respective calibrated MODIS VCF values.**

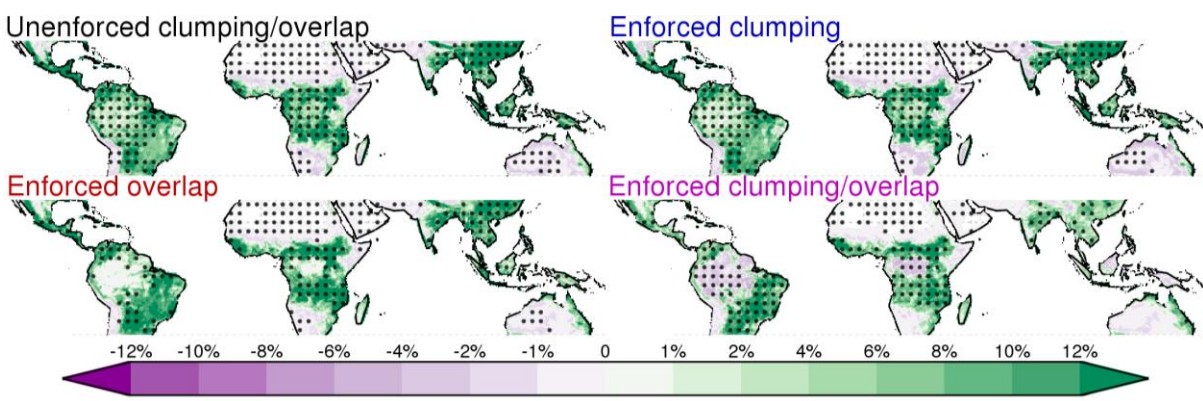

**Figure A2:** The change in tree cover post-calibration for all four scenarios. Black dots indicate areas where the post-calibration values have a 95% certainty of being positive (increasing cover) or negative (decreasing cover) calibrations. These uncertainty maps are indicators of areas where MODIS VCF estimates may be more or less reliable, and cannot be used as definitive calibrations due to the limited number of field sites used as reference.

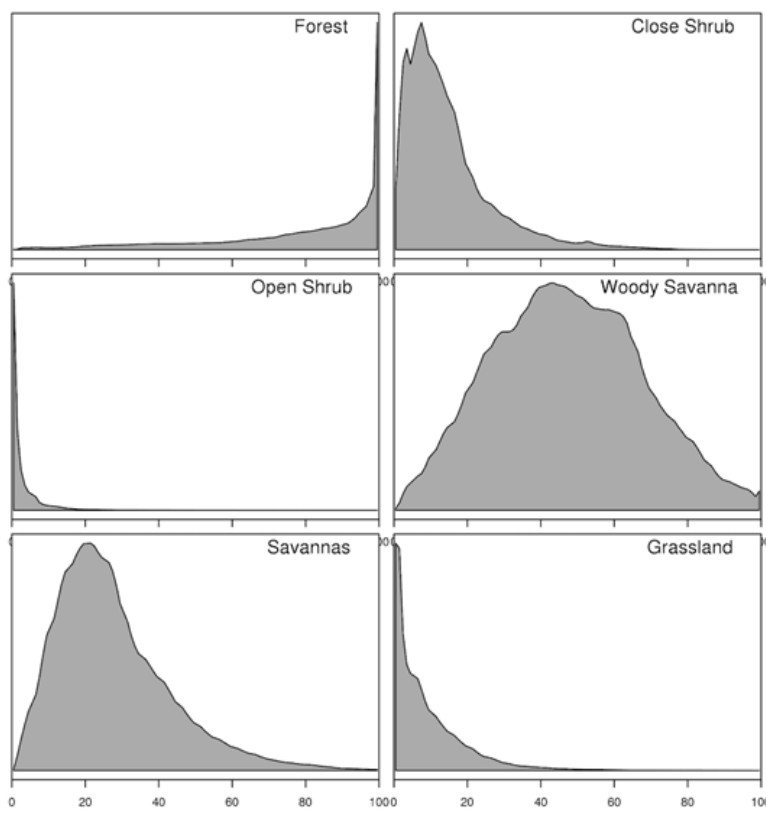


**Figure A3. Frequency distributions of percent tree cover value as estimated by MODIS VCF across the 'forest' supercategory and the following IGBP classes that by our definition count as part of the 'savanna' domain: Closed Shrublands, Open Shrublands, Woody Savannas, Savannas, and Grasslands. Specific class definitions as per the User Guide for the MODIS Land Cover Product (Sulla-Menashe and Friedl, 2018).**


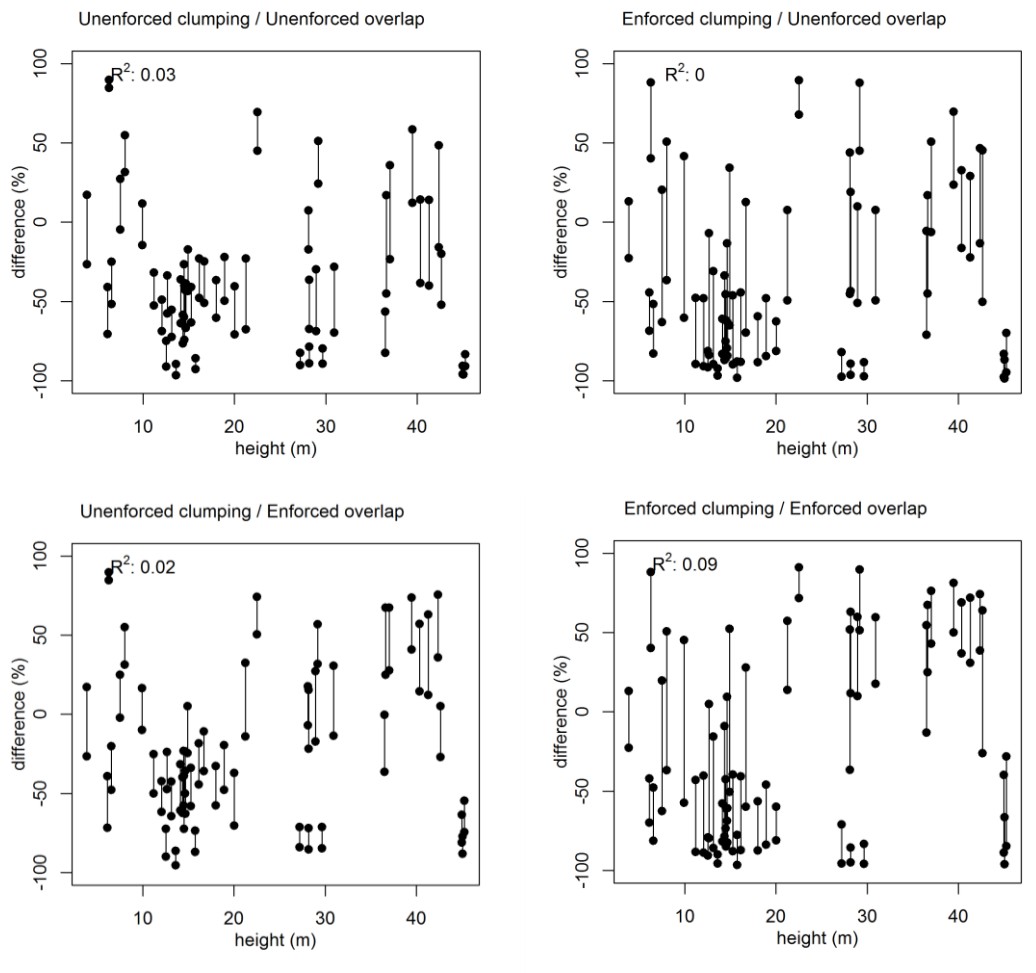

**Figure A4: TROBIT plot upper stratum height versus the difference between MODIS VCF and TROBIT percent tree cover for the four clumping and overlap scenarios. Upper and lower bars represent the uncertainty range's 10th and 90th percentile respectively, based on the convolution of MODIS VCF and TROBIT cover uncertainties from Fig. 4.**


**Code/Data Availability.**

The code and data used to support the findings of this study are archived at
https://github.com/douglask3/VCF_vs_sites revision number fdda3ff

**Author Contribution**.

R.A., D.K., F.G. designed the TROBIT, MODIS VCF pixel-site comparison technique. R.A., F.G. collated
TROBIT and corresponding MODIS VCF values and C.G. collated Sexton et al. (2013) values. D.K. and N.D.

performed regression analysis and constructed global maps. R.A. wrote the first draft of the paper with input
from D.K. and F.G. D.K. plotted the figures. M.T.R, E.V., T.R.F., O.L.P., S.L., B.S., H.T.., B.S.M., T.D., L.A.,
G.D., and J.K. carried out the extensive TROBIT field campaigns. M.T.R. was the main person responsible for
field data quality checking and digitising. R.A., D.K., F.G. and N.D. contributed to the final manuscript.

**Competing Interests.**

The authors declare that they have no conflict of interest.

**Acknowledgements.**

The contribution by D.K. was supported by the U.K. Natural Environment Research Council through The U.K.
Earth System Modelling Project (UKESM, grant no. NE/N017951/1), F.G. was supported by the SUNRISE
project (grant no. NE/R000131/1). N.D. was supported by the Australia Research Council (DP170103410).

B.S.M. was supported by the National Council for Scientific and Technological Development (CNPq,
301153/2018-3).

Maps in Fig. 1 and Fig. 3 were constructed using raster2.6-7 (Hijmans, 2017) and mapproj1.2-5 (Brownrigg et
al., 2017) in R 3.2.0 (R Core Team, 2015). Coastlines were obtained from mapsv3.1.0 (Becker et al., 2016).

Special thanks to Prof. Jon Lloyd for supervising R.A. at the start of the project, to Jeanette Kemp for her work

in the Australian field sites, and to Azemiyah Abdul Rahim for her support during the production of this
manuscript.

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
