# Peer review of "MODIS Vegetation Continuous Fields tree cover needs calibrating in tropical savannas."

_Biogeosciences, 2020_

## Author Comment (AC1)

Thank you to reviewer #1 for your very helpful review. The suggestion to further consider the uncertainty associated with our field data led to additional analysis that strengthens our original point, and we have outlined the additional work and changes made in this response. Reviewer comments are in italics, responses are in blue. Line numbers are from the original manuscript.

*General comments:*

*Bias in MODIS VCF is a known issue. The study by Adzhar et al. adds new interpretation to the issue by comparing the latest version (Collection 6) of the satellite-based dataset with field data collected at ecological sites (24 forest and 24 savanna sites). The authors developed a simulation technique to address the scale discrepancy between the 100m x 100m sites and the 250m x 250m MODIS pixels, and extrapolated the derived relationship across the tropics. The study went on and analyzed the bias patterns of MODIS VCF per land-cover class using the MODIS land cover product. A major finding of the study is that MODIS VCF underestimates tree cover in tropical savannas and woody savannas, which has important implications for carbon cycle and forest restoration potentials. The paper is well written and easy to follow.*

Specific comments:

*Multiple definitions of "tree cover" exist in the literature. The manuscript provides a clear definition of Canopy Area Index (CAI, the fraction of ground covered by tree crowns). A primary semantic difference between CAI, and percent tree cover of MODIS VCF (the portion of skylight orthogonal to the surface that is intercepted by trees), is whether within-crown gaps are taken into account (CAI) or not (VCF). Within-crown gaps can be non-linear and species specific. For example, a recent analysis by Tang et al. (2019) shows that within-crown gap in a 30m pixel increases as tree cover increases and reaches a maximum of ~10% for conifer trees. The current study addressed the definitional discrepancy by dividing MODIS VCF by 0.8 based on Hansen et al. (2002). A discussion on the uncertainties associated with applying this uniform relationship will benefit the manuscript. Better yet, quantitatively analyze the uncertainty with the collected in situ data, if possible.*

The within-canopy gap fraction correction factor (CGF) can vary with vegetation type (Lloyd et al., 2008; 0.34-0.60)  and change with crown cover (Tang et al., 2019: 0.96 - 0.7).

To test how sensitive the results are to our assumption of a constant CGF value, we incorporated a cover (C) dependent CGF term into equation (1). We define CGF as:

$$CGF(C) = k\, C^{1-\rho} \quad \text{where } 0 \text{ and } 0 < k \leq 1 \tag{r1}$$

CGF is a constant value of k when $\rho = 1$. Therefore, the 0.8  from Hansen et al. (2002) uses k = 0.8 and rou = 1. When $\rho < 1$ increases CGF at smaller covers.

 Applying variable CGF to VCF in equation 1:

$$\text{logit}(VCF/k) = C_0 + \log(C^{1/(1-C^2)}) \tag{r2}$$

When we apply the constant CGF of 0.8 in the m/s, we do so before the clumping conversion of VCF to trobit size grid. To make our results comparable, we normalised VCF using the 0.8 value before analysis.

[Figure]

**Figure r1.** The correction curves as per Fig. A5 using equation r2 in place of equation 1.

The large scale patterns of our VCF correction curve when variable CGF is applied (Fig. r1), remain unchanged compared to when the CGF was a constant value (Fig. A4 in the manuscript). VCF mainly underestimates cover regardless of clumping or overlap except for low and high covers where VCF may overestimate tree cover. The transition between under and overestimating does take more extreme values, and this test suggests that assuming constant CGF provides a conservative estimate for the overall correction required for VCF. Therefore, incorporating varying CGF does not alter our main conclusion that VCF significantly underestimates tree cover in savanna-like systems. Also, Lloyd et al. (2008) found no universal relationship between CGF-crown cover and Tang et al. (2019) was conducted in a conifer-dominated landscape. We therefore stick with constant CGF when extrapolating VCF mismatch across the tropics in the main m/s as this map is only meant to indicate likely regions where VCF may require more detailed study, and now include a description of CGF as a potential cause of uncertainty in the discussion:

"...different in-situ measurement techniques tend to measure different types of tree cover (e.g. Fiala et al., 2006; Korhonen et al., 2006; Rautiainen et al., 2005) and each will require a conversion to enable direct comparison with MODIS VCF. In our case, to account for gaps between tree crowns, we applied the 0.8 'gap correction factor' to the CAI. However, on a plot by plot basis the GCF and resulting tree cover could vary widely (Lloyd et al., 2008). With further in-situ data that describes tropical vegetation type-specific GCF variation, we may be able to incorporate site-specific GCFs into our analysis."

*The manuscript can also benefit by adding a clear description of how Canopy Area Index (CAI) is measured or calculated. The reference of Torello-Raventos et al. (2013) is provided,*

*but the details are not sufficient for understanding how CAI is derived and what's the uncertainties associated with CAI at the site level, especially for those who are not familiar with this paper. From Torello-Raventos et al. (2013), it appears that CAI was not directly measured at sites, but was calculated using allometric equations. Then, what is the uncertainty level associated with CAI estimations at the 100 m x 100 m sites, attributable to the allometric equations or the reference data underlying the allometric equations? Site-level uncertainty could be associated with specific direct measurement techniques. Moreover, tree cover definitions and measurement techniques could be connected, as hemispherical photography, terrestrial laser scanner, airborne waveform and discrete-return lidar all tend to measure different types of tree cover (e.g. Fiala et al. 2006, Korhonen et al. 2006, Rautiainen et al. 2005, Tang et al. 2019).*

CAI: We have included and updated text, further explaining how CAI was derived:

CAI is defined as the sum of the projected areas of individual tree canopies divided by the ground area. In the TROBIT project, plot-CAI was derived as follows (Torello-Raventos et al.,2013; Veenendal et al., 2015): Individual tree projected crown area was determined using the average of crown radii measured along the four cardinal points (i.e. from the centre of the stem to the distance furthest from the stem). The plot-wide CAI value is made up of the sum of the upper-stratum, mid-stratum, and subordinate-stratum crown areas. Membership to a stratum is determined by the tree's dbh (upper-stratum: dbh > 10 cm, mid-stratum: 2.5 cm < dbh < 10 cm, and subordinate-stratum: dbh < 2.5 cm, height > 1.5 m). About 50 trees per stratum per plot were measured to derive plot-specific allometric relations between stem diameter and crown area (supplement B of Torello Raventos et al. (2013)). These were then applied to the whole plot to establish plot-level CAI.

Site level uncertainty: Our Bayesian optimization approach accounts for potential errors in TROBIT cover, which includes those caused by allometric construction of CAI, provided errors are unbiased and roughly consistent across sites (Gelman et al., 2013; Kelley et al. 2021). Systematic errors may affect our results, though our correction curves remain robust even if systematic errors in canopy areas were an order of magnitude larger than the range of errors reported in Supplementary Information B of Torello-Raventos et al. (2013) (Fig. r2).

[Figure]

**Fig r2.** Correction curves as per Fig. A4, but with systematic (left) underestimation and (right) overestimations of 10% in the canopy area index across all TROBIT sites.

Also, text has been added in the discussions to highlight the site-level tree cover uncertainty associated with measurement techniques.

*The inherent scale discrepancy between 100 m x 100m sites and 250m x 250m pixels is nicely addressed by simulations. Comparison results between the four types of simulations are also interesting. The authors could consider including the Landsat VCF data (Sexton et al. 2013) in the analysis, which is a satellite-based product most close to MODIS VCF. With a 30m x 30m spatial resolution, Landsat VCF can be averaged to close to the site scale, and a circa-2005 Landsat VCF product is available. This might generate additional insights, and might help resolve the difference with Brandt et al. 2020 in the Sahel region.*

We chose to use the MODIS VCF product specifically because it's one of the most widely used products in climatic and vegetation modelling, and has been for many years (see examples of studies in the introduction). While the 30 m GLCF product introduced by Sexton et al., 2013 is on a scale much more easily comparable to our field data, there are only 2 maps available (for 2000 and 2005), neither of which covers the field campaign period for the TROBIT project. In ecosystems as dynamic as tropical savannas, addressing the temporal mismatch would be very challenging.

In addition, performing the analysis we carried out in this paper on the GLCF product would take approximately 70 times longer than it did for MODIS VCF, rendering this correction technique computationally unfeasible for use on this much smaller scale.

*The striking difference between open shrub and savannas (Figure 3) is puzzling, even with the discussion provided. Much like fractional land cover (e.g. MODIS VCF), the accuracy of discrete land cover classification such as the MODIS IGBP land cover product is also generally lower over open canopy ecosystems, and misclassifications often occur among*

*those classes. Could the in situ data provide some estimates on the accuracy of MODIS land cover product as an additional analysis? In addition, if open shrubland and grassland do have higher accuracy, for completeness, this might be better pointed out in the abstract.*

The reviewer is correct to highlight classification accuracy as a possible explanation for the observed difference between open shrub and savannas in Figure 3. The MODIS IGBP confusion matrix (Table S6 in Sulla-Menashe et al., 2019) shows that the majority of the 'open shrubland' class' commission error is with the 'grassland' class. MODIS VCF tree cover in areas classified as 'open shrublands' is therefore likely to be lower than expected from the IGBP definition. The matrix also shows a high commission error ($> 0.7$) for 'savannas' involving all IGBP classes considered in our study and very high commission and omission errors ($> 0.9$) for 'closed shrublands'. We have rewritten relevant parts of our discussion section to focus on accuracy:

"...we found discrepancies between the tree cover ranges in the IGBP class definitions and the class specific ranges estimated through MODIS VCF (Fig. A5), which suggests that the 5 m height threshold may not always apply in MODIS VCF. There is also no clear relationship between TROBITs upper stratum canopy height and the difference in TROBIT and VCF covers (Fig. A6). In this case, the inclusion of trees below this 5 m height in the TROBIT inventory would not fully explain this underestimation. For example, MODIS VCF recorded tree cover in the 'open shrublands' and 'closed shrublands' classes of the MCD12Q1 product (Fig. A5), even though the height range for these classes is 1 - 2 m. For the IGBP 'savannas' class, MODIS VCF yields a percent tree cover range that matches closely with the 'savannas' class definition (between 10 % and 30 %), despite the differing tree thresholds for MODIS VCF and IGBP (5 m minimum for MODIS VCF, and 2 m minimum for IGBP) are different. These discrepancies suggest one of the following three things: 'open/closed shrublands' and 'savannas' contain trees taller than 5 m; MODIS VCF is distinguishing trees below the 5 m threshold; or, some combination of both.

The accuracy of the MCD12Q1 product is also class dependent (Fig. 4). The accuracy for 'closed shrublands' is particularly low. It is mainly confused with open shrublands, woody savannas and savannas. There is also confusion to a lesser extent between open shrubland, woody savannas, savannas, and grasslands. These between class omission and commission errors could be another explanation for the discrepancy between the IGBP class tree cover ranges and those estimated through MODIS VCF.

More work needs to be done to evaluate how effective both MODIS VCF and MCD12Q1 are at implementing the height thresholds in their respective 'tree' definitions, as this may have implications when MODIS VCF and MCD12Q1 (and their tree definitions) are used for global model calibration or validation."

We mainly used the IGBP product to get a general sense of how MODIS VCF behaves across different cover types, and the product is reliable enough for this purpose. How well the IGBP product classifies land cover matching its class definitions is certainly something that requires further examination, and using in-situ data (such as our TROBIT dataset) as a reference to assess it would be something to consider for further study. However, this would require a different statistical approach to the one developed here.

*Technical corrections:*

*Figure 1. Quite a few savanna sites have TROBIT tree cover > 60%, which falls in the definition of forest in the text (lines 22-23). Are those sites better considered as savanna or treated as forest sites?*

Owing to the enormous structural variation that can be found in savannas, we use 'mean tree height' as a major distinguishing factor over tree cover to delineate forest from savanna. As per Veenendaal et al. (2015) and Torello-Raventos et al. (2013), a stand needs to have both tree cover exceeding 60 % and a mean tree height of or exceeding 6 m to qualify as a forest, as per lines 120 -125.  Many modelling studies also delineate forest and savanna by height (e.g. Prentice et al., 2011; Sato et al., 2021; Martin Calvo, 2015; Kelley et al. 2013). The plots classified as 'savanna' as per the TROBIT project can therefore have a tree cover > 60 % and still not count as 'forest'.

Line 313, change "classed" to "classified"

We have changed line 313 in the revised manuscript.

References:
Gelman, A., Carlin, J. B., Stern, H. S., Dunson, D. B., Vehtari, A. and Rubin, D. B.: Bayesian Data Analysis, Third Edition, CRC Press., 2013.

Kelley, D. I., Prentice, I. C., Harrison, S. P., Wang, H., Simard, M., Fisher, J. B. and Willis, K. O.: A comprehensive benchmarking system for evaluating global vegetation models, Biogeosciences, 10(5), 3313–3340, doi:https://doi.org/10.5194/bg-10-3313-2013, 2013.

Kelley, D. I., Burton, C., Huntingford, C., Brown, M. A. J., Whitley, R., and Dong, N.: Technical note: Low meteorological influence found in 2019 Amazonia fires, Biogeosciences, 18, 787–804, https://doi.org/10.5194/bg-18-787-2021, 2021.

Lloyd, J., Bird, M. I., Vellen, L., Miranda, A. C., Veenendaal, E. M., Djagbletey, G., Miranda, H. S., Cook, G. and Farquhar, G. D.: Contributions of woody and herbaceous vegetation to tropical savanna ecosystem productivity: a quasi-global estimate, Tree Physiology, 28(3), 451–468, doi:10.1093/treephys/28.3.451, 2008.

Martin Calvo, M. & Prentice, I. C. Effects of fire and CO2 on biogeography and primary production in glacial and modern climates. New Phytol. 208, 987–994, 2015.

Prentice, I. C., S. P. Harrison, and P. J. Bartlein. "Global vegetation and terrestrial carbon cycle changes after the last ice age." New Phytologist 189(4), 988-998, 2011.

Sato H, Kelley DO, Mayor SJ, Calvo MM, Cowling SA, Prentice IC, "Dry corridors opened by fire and low CO2 in Amazonian rainforest during the Last Glacial Maximum", Nature Geoscience, 2021.

Sulla-Menashe, D., Gray, J. M., Abercrombie, S. P. and Friedl, M. A.: Hierarchical mapping of annual global land cover 2001 to present: The MODIS Collection 6 Land Cover product, Remote Sensing of Environment, 222, 183-194, doi:10.1016/j.rse.2018.12.013, 2019.

Sexton, J. O., Song, X.-P., Feng, M., Noojipady, P., Anand, A., Huang, C., Kim, D.-H., Collins, K. M., Channan, S., DiMiceli, C. and Townshend, J. R.: Global, 30-m resolution continuous fields of tree cover: Landsat-based rescaling of MODIS vegetation continuous fields with lidar-based estimates of error, International Journal of Digital Earth, 6(5), 427–448, doi:10.1080/17538947.2013.786146, 2013.

Tang, H., Song, X.-P., Zhao, F. A., Strahler, A. H., Schaaf, C. L., Goetz, S., Huang, C., Hansen, M. C. and Dubayah, R.: Definition and measurement of tree cover: A comparative analysis of field-, lidar- and landsat-based tree cover estimations in the Sierra national forests, USA, Agricultural and Forest Meteorology, 268, 258-268, doi: 10.1016/j.agrformet.2019.01.024, 2019.

Torello-Raventos, M., Feldpausch, T., Veenendaal, E., Schrodt, F., Saiz, G., Domingues, T., Djagbletey, G., Ford, A., Kemp, J., Marimon, B., Marimon-Junior, B. H., Lenza, E., A Ratter, J., Maracahipes, L., Sasaki, D., Sonké, B., Zapfack, L., Taedoumg, H., Daniel, V. and Lloyd, J.: On the delineation of tropical vegetation types with an emphasis on forest/savanna transitions, Plant Ecology & Diversity, 6(1), 101–137, doi:10.1080/17550874.2012.762812, 2013.

Veenendaal, E. M., Torello-Raventos, M., Feldpausch, T. R., Domingues, T. F., Gerard, F., Schrodt, F., Saiz, G., Quesada, C. A., Djagbletey, G., Ford, A., Kemp, J., Marimon, B. S., Marimon-Junior, B. H., Lenza, E., Ratter, J. A., Maracahipes, L., Sasaki, D., Sonké, B., Zapfack, L., Villarroel, D., Schwarz, M., Yoko Ishida, F., Gilpin, M., Nardoto, G. B., Affum-Baffoe, K., Arroyo, L., Bloomfield, K., Ceca, G., Compaore, H., Davies, K., Diallo, A., Fyllas, N. M., Gignoux, J., Hien, F., Johnson, M., Mougin, E., Hiernaux, P., Killeen, T., Metcalfe, D., Miranda, H. S., Steininger, M., Sykora, K., Bird, M. I., Grace, J., Lewis, S., Phillips, O. L. and Lloyd, J.: Structural, physiognomic and above-ground biomass variation in savanna–forest transition zones on three continents – how different are co-occurring savanna and forest formations?, Biogeosciences, 12(10), 2927–2951, doi:10.5194/bg-12-2927-2015, 2015.

---

## Author Comment (AC2)

Thank you to reviewer #2 for your insightful review. In response to your concerns about applying our results across the tropics and discussing height thresholds, we have expanded our Discussion section to clarify the strengths and weaknesses of our approach in mapping uncertainty across the tropics, and added additional relevant information regarding plot heights. Reviewer comments are in italics, responses are in blue. Line numbers are from the original manuscript.

*This manuscript undertakes an analysis of the accuracy of the MODIS Vegetation Continuous Fields product with particular reference to sparsely wooded ecosystems using tropical forest and savanna field inventory data.*

*The manuscript is well written and presented. The figures are of high quality and the analysis is clearly described. As the authors related the VCF has been subject to considerable analysis and discussion over some period of time. It could reasonable be postulated that all products from moderate resolution sensors struggle with accurate discrimination along the ecotone between forest and grassland due to non-linearilties in the reflectance and VIs applied to derive them, and due to the enormous variation in the morphology, architecture, density and clumping, and phenology of the overstory.*

*The authors have to work with a limited inventory data set - limited in geographical coverage, and limited in sampling the above mentioned variation. They quite reasonably focus on the issue of clumping of the tree fraction within pixels and undertake a nice analysis based on this. I do wonder why they did not explore the actual representativeness of the field inventory sites versus the VCF pixel resolution in the manner of Roman et al., 2009 and subsequent publications.*

*[M.O. Román, C.B. Schaaf, C.E. Woodcock, A.H. Strahler, X. Yang, R.H. Braswell, P.S. Curtis, K.J. Davis, D. Dragoni, M.L. Goulden The MODIS (collection V005) BRDF/albedo product: assessment of spatial representativeness over forested landscapes Remote Sens. Environ., 113 (2009), pp. 2476-2498, 10.1016/j.rse.2009.07.009]*

*This is not in any way disqualifying since the analysis here is coherent and valid in itself.*

We chose to use 'clumping' and 'overlap' parameters to scale the field sites up to MODIS VCF pixel sizes because it allows us to simulate a huge variety of tree cover distribution, with enormous potential variation between the site and the pixel. We find that this is a good way to fully consider the very different potential structural compositions of savannas, particularly those within ecotones. The approach of Román et la. (2009) is an interesting technique that could be used to constrain uncertainty further by measuring the coefficient of variation between the site and pixel, and would be something to consider in future research.

*However, I think that there are several issues that the authors need to address more fully especially in their Results and Discussion.*

*1. The manuscript proposes a correction to VCF for savannas and forests and combined. The authors state in the Results (lines 224-229) that VCF estimates forest tree cover well and greatly under-estimates savanna tree cover. However, based on Figure 1. the performance of VCF at the forest sites looks pretty terrible based on the sites being in a pretty clear blob with very wide variation between Trobit and VCF. This makes me wonder about the representativeness issue for these Trobit sites and VCF pixels and whether clumping issues*

*are the only thing happening here. In any case, Figure 1 does not suggest that VCF is doing well in forest but is this a sampling issue?*

On a site-by-site basis it is true that there are many forest points that do not appear to be accurately measured by MODIS VCF as seen by their distance to the 1:1 line. The main difference we want to highlight is that MODIS VCF behaves very differently in forests versus savannas: where in savannas it consistently overestimates, in forests it behaves much less consistently, both over- and under-estimating cover. The lack of significant tree cover gain or loss for forests, after correction, across all combinations of clumping/overlap (Fig. 3) further demonstrates this. We have edited the text in section 3.1 of the Results to clarify that MODIS VCF does not perform 'well' in the TROBIT ecotone forests, but it does measure cover without the systematic offset observed in the savannas, which is our focus in this paper.

We have also inserted text in the Methods section to clarify that the forest and savanna sites in TROBIT were selected because they represent the forest-savanna ecotone, and so while MODIS VCF appears to have some issues with TROBIT forests, this does not necessarily hold true for other forest types. Further work looking at the product's effectiveness across different forest types may yield greater insight into how MODIS VCF performs in case of  denser forest types.

*2. I am concerned about proposing an overall correction to VCF based on such a limited sampling of the ecotone between grassland and forest. I wonder if it would be better to more clearly identify in the written text, the kind of savanna that is sampled by the Trobit.*

The figure is designed to show potential areas of VCF mis-estimates rather than a correction. Note that we only show difference and significance maps - not an overall VCF correction map. This is important to help inform future targeted VCF assessments. We have emphasised (within the description of Fig. 2) how our post-correction maps are meant to be used: as indicators for where the product may be more or less reliable.

*Figure A1 shows the distribution of sites. Although the sites appear to sample the gradient between the amazon and the cerrado reasonably well (however there is enormous variation within cerrado from wooded to very open short sparse shrubland), they do not sample the variation in savanna structure in Africa and Australia very well. The African sampling is confined to the tree cover gradient in West Africa passing from the Guinean Savana to the Sahel, whilst the sampling in Australia is confined to areas around the small tropical rainforest area and surrounding Einasleigh Uplands. in northern Queensland. There is enormous variation in structure, morphology, and phenology across African and Australian savannas. I would find the study more compelling if it: 1) paid more attention to the actual species, structure , phenology and composition of the sites; 2) constrained the narrative to an analysis of VCF for these particular systems; and 3) suggested a correction approach that is relevant to these kinds of systems and maintains the focus on the issue of sparsely wooded systems. I believe that it is a stretch to propose an overal correction to VCF from this study simply becuse of the available inventory data are limited in coverage of geographical and vegetation diversity.*

We chose to do a tropic wide analysis as it is the ecological modelling studies carried out at these scales that often use VCF without considering VCF's underlying uncertainties (see references in the introduction, and discussion in the original m/s).  The main purpose of our

paper is to highlight to the global vegetation modelling community that VCF may have significant and systematic errors even when you take a broad view, and to encourage caution in the use of VCF. The reviewer and our introduction provide references where local-scale VCF assessments have been conducted, but none have translated their results demonstrating tropics-wide spatial implications.

However, the reviewer is rightly concerned that our analysis could be construed as a usable "correction map" of tropics-wide VCF. We placed several safeguards against this in the original m/s:

1. We only show difference maps, and no definitive "correction map"
2. Figure 2 maps are for all four clumping/overlap scenarios, and although these show qualitatively similar general patterns, there are also significant quantitative differences between them.
3. Disagreement between these maps are highlighted by the "Significance" map in Fig. 2
4. Much of the discussion is dedicated to constraining these uncertainties between scenarios further.

In addition, in the revised m/s, we make a number of changes and additions to make clear that we are presenting "areas of concern" when using VCF and not definitive corrected VCF maps:

1. We removed any reference to "corrected VCF" when discussing tropics-wide maps.
2. In the discussion, we included a map (Figure r2.1 - to be included as Fig A6) showing where to prioritise future in-situ reference data collection to help constrain our uncertainty maps further. We also included the following text in the discussion:

   "Finally, using a limited number of field plots will create additional uncertainty when the calibration is applied across the tropics. To identify where additional field plots would reduce this uncertainty we combined our uncertainty maps (Fig. 2) with distance from TROBIT plots (See Fig A6 in appendix). "

3.      Additionally, as the reviewer points out, vegetation structure, morphology and phenology are all variables that could make a difference. Yet, although inter- and intra-continental differences in these variables between TROBIT plots are already quite large (Torello-Raventos et al. (2013) and Veenendaal et al. 2015), all continents show similar patterns across our clumping/overlap scenarios for savanna, i.e. a smaller underestimation or slight overestimation of tree cover at small and large covers and a substantial underestimation at intermediate covers (Fig. r2.2). To acknowledge the impact of plot representativeness (within the highly variable global tropical forest-savanna ecotone) on our analysis we have added the following text in our Discussion:

   "Using a limited number of field plots will create additional uncertainty when the calibration is applied globally across this highly variable tropical forest-savanna ecotone. The map in Fig. A6 combines our uncertainty maps (Fig. 2) with distance from TROBIT plots, and highlights Southeast Asia, Central America, and Mexico as areas where additional in-situ tree cover observations would help constrain uncertainties. Field data from the northwestern region of South America, the southeast of the African continent, and Madagascar would also help. Finally, factors such as topography, soil type and moisture content, phenology and cloud cover, and landscape heterogeneity, can affect the accuracy of remotely-sensed products, including MODIS

VCF. Data, characterising these at plot level, would help identify potential confounding factors affecting MODIS VCF performance and so help constrain uncertainties further."

[Figure]

**Fig r2.1.** (Top) uncertainty range of potential VCF mismatch, calculated as maximum 90% percentile minus minimum 10% quantile over the four scenarios in Fig. 2. (Middle) geographic distance to closest TROBIT site. (Bottom) based on top and middle maps, priority areas for additional data collection to further constrain map uncertainties of Fig. 2.

[Figure]

**Fig r2.2:** TROBIT vs VCF as per Fig 1 in the m/s, but with points coloured by continent: South America in green, Africa in red, Australia in purple. Triangles are savanna, circles are forests. Greyed regression polygons match the coloured forest, savanna and overall regression cures in Fig. 1.

*3. A key issue with VCF and this analysis is the thorny one of the definition of tree cover. Really with VCF and any other remote sensing products it should be about detection of woody cover (of any sort) from the canopy reflectance which is distinguished from the background soil and understory wherein the sensitivity is constrained by pixel resolution and the discriminatory capacity of available reflectance bands. I think that the manuscript gets a bit bogged down in this thorny issue since VCF has a certain definition which is apparently based on height. This is problematice in many ways since although related, height does not have a one to one relationship with canopy extent. However this MODIS product has been around for a long time with clear definition (of height). This rather means that analysis using these Trobit sites needs to provide information on the "tree height" distributions at these sites. This once again returns to representativeness, and maybe explains why the forest data are so scattered for Trobit versis VCF in Figure 1 . So again I want more information about what the vegetation is at these Trobit sites. As a result, the Discussion is a bit convoluted. The section between lines 301 and 319 therefore is rather confusing and muddled to read. A whole lot of speculation about where trees are > 5 m or not is included. To make statements about this one needs evidence of the tree height distribution at the Trobit sites and a matching analysis. I treat VCF as attempting to estimate woody cover. I don't care if it is getting at trees > 5 m or less.*

*So this is kind of a furphy. The results in Figure 1 for savannas are pretty convincing about underestimation of woody cover which is probably unsurprising given the VCF method, but readily corrected based on your analysis at least for these particular savanna systems. If the Trobit data are limited then a whole lot of speculation about tree height really is not relevant and just clouds the Discussion. It damages the message which is best from the savanna analysis. Hence I think that clearer and more simplified findings and discussion points are required.*

We agree with the reviewer that for RS-based products, such as VCF and MCD12Q1, that are derived from optical sensor data (i.e. surface reflectance), implementing a height threshold will be difficult. However, this height threshold is clearly stated in the VCF's definition and, similarly to the % cover thresholds in the IGBP land cover class definitions, is taken at face value by field-based scientists and modellers. As a matter of fact, model outputs are often adapted to match this 5 m threshold (e.g. Kelley et al., 2013; Lasslop et al., 2018). It is therefore important we discuss height. And while reviewer #2 is correct in saying that we cannot conclusively prove that height is a factor at play with MODIS VCF's accuracy with the limited data at hand, we find it important to highlight the importance of caution when assuming the 5 m limit.

However, we also agree that there was room for improvement in the manner in which we tackled this thorny issue and have improved our text in the Results and Discussion part of the manuscript. We have now provided the TROBIT plot heights and cover type by Torello-Raventos et al. (2013) in Table A1. Using this height information, we have also shown there is no relationship between TROBIT height and the difference between VCF and TROBIT covers (Fig r2.3 - to be included at Fig. A7 in the revised m/s). While this does not definitively disprove the existence of an assumed height threshold, the fact that there is no reduction in error as the upper stratum exceeds 5m warrants further investigation into the 5m limit.

[Figure]

**Fig r.2.3**: TROBIT upper stratum height vs difference between VCF and TROBIT % cover over our four clumping and overlap scenarios. Upper and low bars represent 10-90 percentiles of uncertainty range based on convolution of VCF and TROBIT covers uncertainties from Fig. 1.

*In summary, the study is well written, the methods are good, and the presentation is excellent. It is an interesting and useful study. However the authors need to address the following major issues before it can be considered for publication.*

*1. Limit the scope to developing correction to VCF that can be applied in the systems where Trobit inventory is available and representative.*

*2. Address the issue of representativeness of the Trobit sites, describe them much more fully, provide information of tree heights or tree height distributions or information from the literature on typical structure for these types of savannas and forests.*

*3. Focus the correction of the savannas for which your analysis is more convincing and explain the variation in the forest tree cover and why you think that VCF is good based on Figure 1.*

*4. Simplify the issues around definitions and detection of "trees". If you have no height data you can't really comment on this. Make sure that the thrust of the Results and Discussion is clear and clean and does not jump around between phrases about height definition when the imagery is seeing tree canopies not heights. If you can better describe the height distributions and representativeness of the sites, then discussion of the height definition can be useful.*

In conclusion, these are the steps we have taken to address your concerns:

1. We will continue to map our findings across the tropics as this scale is one that best informs work done using VCF in ecological modelling, but we have made our limitations very clear and have emphasised that our findings are a guide to where there is likely more or less uncertainty, and not a correction within itself.
2. Additional information regarding TROBIT plots have been added to the paper, describing the types of plots selected for the project on the whole, with relevant plot-specific details added to table A1.
3. We have clarified that VCF use in savannas is more concerning than in forests because of a persistent systematic offset, and better described how VCF performs in forests in the Results section.
4. The Discussions section has been re-worked to better explain that the assumed height threshold in VCF has not been reflected in our results, and why it is an important issue to discuss.

References:

Kelley, D. I., Prentice, I. C., Harrison, S. P., Wang, H., Simard, M., Fisher, J. B. and Willis, K. O.: A comprehensive benchmarking system for evaluating global vegetation models, Biogeosciences, 10(5), 3313–3340, doi:https://doi.org/10.5194/bg-10-3313-2013, 2013.

Lasslop, G., Moeller, T., D'Onofrio, D., Hantson, S. and Kloster, S.: Tropical climate–vegetation–fire relationships: multivariate evaluation of the land surface model JSBACH, Biogeosciences, 15(19), 5969–5989, doi:https://doi.org/10.5194/bg-15-5969-2018, 2018.

Torello-Raventos, M., Feldpausch, T., Veenendaal, E., Schrodt, F., Saiz, G., Domingues, T., Djagbletey, G., Ford, A., Kemp, J., Marimon, B., Marimon-Junior, B. H., Lenza, E., A Ratter, J., Maracahipes, L., Sasaki, D., Sonké, B., Zapfack, L., Taedoumg, H., Daniel, V. and Lloyd, J.: On the delineation of tropical vegetation types with an emphasis on forest/savanna transitions, Plant Ecology & Diversity, 6(1), 101–137, doi:10.1080/17550874.2012.762812, 2013.

Veenendaal, E. M., Torello-Raventos, M., Feldpausch, T. R., Domingues, T. F., Gerard, F., Schrodt, F., Saiz, G., Quesada, C. A., Djagbletey, G., Ford, A., Kemp, J., Marimon, B. S., Marimon-Junior, B. H., Lenza, E., Ratter, J. A., Maracahipes, L., Sasaki, D., Sonké, B., Zapfack, L., Villarroel, D., Schwarz, M., Yoko Ishida, F., Gilpin, M., Nardoto, G. B., Affum-Baffoe, K., Arroyo, L., Bloomfield, K., Ceca, G., Compaore, H., Davies, K., Diallo, A., Fyllas, N. M., Gignoux, J., Hien, F., Johnson, M., Mougin, E., Hiernaux, P., Killeen, T., Metcalfe, D., Miranda, H. S., Steininger, M., Sykora, K., Bird, M. I., Grace, J., Lewis, S.,

Phillips, O. L. and Lloyd, J.: Structural, physiognomic and above-ground biomass variation in savanna–forest transition zones on three continents – how different are co-occurring savanna and forest formations?, Biogeosciences, 12(10), 2927–2951, doi:10.5194/bg-12-2927-2015, 2015.

---

## Author Comment (AC3)

Thank you to reviewer #3 for your informative review. In response to your concerns about applying our results across the tropics, we have expanded our Discussion section to clarify our reasoning for creating a tropics-wide uncertainty map, and included additional information regarding the sites. Reviewer comments are in italics, responses are in blue. Line numbers are from the original manuscript.

*This study evaluates the bias of MODIS VCF with field data and implements a correction for VCF. The manuscript is well written and presented with nice figures. But I am more interested in the method part you developed to do comparisons than the corrected VCF map and the findings of underestimated tree cover which are not surprising. As I get worried about the feasibility of using the limited number of samples to correct MODIS VCF maps covering the whole tropics, which will further make the related results pale. For the IGBP map, please specify year used and the spatial coverage of IGBP for grouping*

IGBP was +/- 30Deg North and year 2006-2009, as per MODIS VCF and field plots. We will include these details in the methods.

*My suggestion is to just focus your analysis at the site level or to limit your study area to where the TROPBIT data are representative.*

This paper is targeted towards the use of the MODIS VCF product in global environmental models, which requires extrapolating the TROBIT-VCF mismatch across the tropics. The focus is to highlight potential problems introduced when VCF is used as a definitive tree cover target in studies using these global modelling approaches. We have made the limitations of using a limited number of field sites and justifications clearer in our Discussion. See response to reviewer 2, point 1 and 2 for more detail.

*Small questions,*

*1, specify the meaning of the terms "tree cover distributions"*

We have corrected the incidences of 'tree cover distributions' to 'tree cover frequency distributions', and provided the definition within the Methods section.

*2, should consider moving the details about the TROBIT field data to methods section*

We have moved the relevant details to Methods.

*3, please specify the rationality of using the equation 1.*

A logit transformation is a standard approach to transform double-bounded variables (in our case, fractional cover - bounded between 0 and 1) for regression. This prevents predictions based on regression from taking unphysical values (i.e covers less than zero or greater than 1) (Gelman et al. 2013; Bistinas et al., 2014; Kelley et al. 2019, 2021). While other transformations are also useful for regression of bounded data, logit maintains data ranking – important as, to our knowledge, there is no indication of systematic ranking issues in either VCF or TROBIT.

Also our Bayesian regression technique assigns the likelihood of a parameter set based on the conditional probability of a VCF value given our VCF reconstruction from TROBIT. As already stated in the methods, the best way to calculate this probability is to ensure a normally distributed regression model. On the right hand side of the equation, we transform

TROBIT cover using a logit-like function with two extra parameters to account for known asymmetry in cover distributions, which may affect the normality of error distribution. We have already detailed this in the methods ("This is similar to a standard linear regression of logit transformed data, accounting for maximum and minimum bounds of 0-100 % tree cover, allowing for a non-symmetric transformation of tree cover." - line 160 -165 in the current iteration of the manuscript).

References:

Bistinas, I., Harrison, S. P., Prentice, I. C., and Pereira, J. M. C.: Causal relationships versus emergent patterns in the global controls of fire frequency, Biogeosciences, 11, 5087–5101, https://doi.org/10.5194/bg-11-5087-2014, 2014.

Gelman, A., Carlin, J. B., Stern, H. S., Dunson, D. B., Vehtari, A. and Rubin, D. B.: Bayesian Data Analysis, Third Edition, CRC Press., 2013.

Kelley, D. I., Bistinas, I., Whitley, R., Burton, C., Marthews, T. R. and Dong, N.: How contemporary bioclimatic and human controls change global fire regimes, Nature Climate Change, 9(9), 690–696, doi:10.1038/s41558-019-0540-7, 2019.

Kelley, D. I., Burton, C., Huntingford, C., Brown, M. A. J., Whitley, R., and Dong, N.: Technical note: Low meteorological influence found in 2019 Amazonia fires, Biogeosciences, 18, 787–804, https://doi.org/10.5194/bg-18-787-2021, 2021.

---

## Author Response (AR1)

We have provided a point-by-point description of the changes to the m/s in our responses to reviewers (see AC1, AC2, and AC3).

On top of the changes outlined in these responses, we have also updated the m/s as follows:
1. Updated affiliations for several co-authors.
2. Included additional details regarding the field sites sampled in the Methods section, and refined our definition for 'forests' and savannas'.
3. Re-run our analysis to better reflect the new 'forest' and 'savanna' categories, as well as to account for different-sized field sites.
4. Updated Table A1 with additional site-specific information, and added figures A6 and A7 to address concerns regarding our limited datasets brought up by Reviewer 2.
5. Corrected mistakes and made grammatical improvements throughout.

We would also like to address the formatting issue in our r1 and r2 equations in our response to AC1. The r1 equation should instead be:

$$GCF(C) = k \times C^{1-\rho}$$

And r2 should instead be:

$$logit(VCF^{\rho}/k) = C_0 + \Delta \times log(C^{\tau_1}/(1 - C^{\tau_2}))$$

When $\rho < 1$, the GCF decreases with decreasing tree cover.

We have included a copy of the manuscript with track changes.

Please let us know if any more information is required.

Thanks

Rahayu Adzhar on behalf of co-authors

---

## Author Response (AR2)

Dear Editor, please find attached our revised manuscript. We have dealt with all the reviewers' comments and your recommendations and hope this improved version has addressed all the outstanding issues.

We have highlighted the limitations of the global extrapolation further by referring to 'calibration' instead of 'correction' throughout the text. We have also replaced Figure 2 with a new figure (Figure 5) that combines Figure 2 and Figure A7 and now shows (i) the post-calibration change in tree cover that is statistically significant across all four scenarios, (ii) the uncertainty range of the calibration, (iii) priority regions for field surveying. The accompanying text in the results and parts of the discussion further stress the uncertainty of our calibration. A detailed outline of the changes in response to the reviewers' comments is below.

We have split the methods section into four sub-sections: 2.1  EO Products and Field data; 2.2 Converting In-Situ Canopy Area Index to MODIS VCF / Landsat TCC percent tree cover;  2.3 Calculating Uncertainty Under Different Overlap-Clumping Scenarios; 2.4 Mapping MODIS VCF Uncertainty Across The Tropics.

We have included appendix figures A2 and A3 (adapted to show the approach used for Landsat TCC) as Figures 1 and 2 in the main text. Figure 2 now also includes an example of the effects of unenforced and enforced clumping on 30 m x 30 m Landsat TCC pixels.

Reviewer1:
Thank you to reviewer #1 for your further review. In response to your concerns about ensuring the reader is aware of how the reference data were collected and in particular the fact that the CAI is derived from allometric relationships, we have split the methods section into 4 sub-section and included the paragraph describing reference data collection in what is now Section 2.2 .  With respect to including the Landsat TCC in our analysis, we are now presenting results of the TROBIT reference data - Landsat TCC tree cover comparison  Please refer to the supplementary pdf for a comprehensive response to your review.

For PDF:
*"...the key steps of reference data collection should still be summarized and disclosed in this paper. It is critical to let the readers beware that the CAI reference used in the research was not directly measured in the field, but was calculated based on allometric equations."*

In what is now Section 2.2 we describe how the reference data is collected and clearly highlight the fact that CAI is based on allometric equations: *"CAI is defined as the sum of the projected areas of individual tree crowns divided by the ground area. In the TROBIT project (Torello-Raventos et al. (2013) and Veenendal et al. (2015)), plot-wide CAI is made up of the sum of the upper-stratum, mid-stratum, and subordinate-stratum crown areas. Membership to a stratum is determined by the tree's dbh (upper-stratum: dbh > 10 cm, mid-stratum: 2.5 cm < dbh < 10 cm, and subordinate-stratum: dbh < 2.5 cm, height > 1.5 m). About 50 trees per stratum per plot were measured to derive plot-specific allometric relations between stem diameter and crown area (supplement B of Torello Raventos et al., 2013). These were then applied to the whole plot to establish plot-level CAI. For the allometric relationships, tree crowns were treated as circles and the individual tree projected crown area was determined using the average of crown radii measured along the four cardinal points (i.e. from the centre of the stem to the distance furthest from the stem). "*

*"...with regards to the suggestion of including the Landsat VCF in additional analysis. I have to rather insist on the comment...such an evaluation can still provide useful insights on how big a role the scale mismatch between the MODIS sensor and field plots plays in the data comparison. It can also provide useful insights on whether or not the biases in MODIS VCF are inherited by the later generation of remote sensing products…"*

We have incorporated the additional Landsat TCC product evaluation into our work. Taking advantage of a recent version4 release which provides a 2005 and 2010 layer, we compared the average of 2005 and 2010 tree cover with the TROBIT reference data.

The text in methods was updated as follows:

*"*We used the 2005 and 2010 30m Landsat TCC version 4 product (https://lcluc.umd.edu/metadata/global-30m-landsat-tree-canopy-version-4), and worked with the 2005 and 2010 average values. The product was downloaded manually from https://e4ftl01.cr.usgs.gov/MEASURES/GFCC30TC.003/. *"*

*"For Landsat TCC, where the Landsat TCC pixels (30m x30m) are smaller than the TROBIT field sites, we calculated a TCC percent tree cover to match the TROBIT field site size by summing the percent tree cover within the TCC pixel part found inside the TROBIT field site and then dividing the sum by the TROBIT site area. As TROBIT site orientation was not recorded, we randomized the angle between the*

*TROBIT site and TCC pixel grid for each of the1000 samples when generating the probability distribution. "Enforced clumping" was performed as per MODIS VCF (Fig. 2)with the direction of clumping randomized. "*

Figure r1, incorporated into the new Figure 2 illustrates the clumping procedure for Landsat TCC.

[Figure]

Figure r1: Example of the effects of unenforced and enforced clumping on 30 m x 30 m Landsat TCC pixels with a mix of tree covers (green) and non-tree cover (brown). White dotted lines are TCC pixel boundaries. Clumping all the cover to one side of the pixel (right bottom) affects the average canopy cover value of a 100 m x 100 m-sized TROBIT site (black boxes).

We found that using MODIS VCF to train the model used to estimate Landsat TCC tree cover propagates the MODIS VCF biases. We have highlighted this is in the text:

*"Similar patterns can be observed with Landsat TCC (black line, Fig. 4). There is a significant underestimation of tree cover in the lower cover ranges up to 59% when there is enforced overlap, and up to 82% when overlap is not enforced. In savanna sites (orange line, Fig. 4) the underestimation (at 95% confidence) is significant and consistent for covers below 75 - 80 % (without enforced overlap) or below 52 - 60% (with enforced overlap). In forest sites (green line, Fig. 4) there is no systematic difference."*

*"While MODIS VCF is a powerful and accessible tool to map tree cover, our field data-based calibrations indicate that the latest MODIS VCF collection 6 is missing a lot of woody cover, even when uncertainty introduced by site canopy overlap and clumping within the MODIS VCF pixel are accounted for. The Landsat TCC product, which may be viewed as an alternative with a higher spatial resolution, behaves in a similar manner."*

*"Similar results for MODIS VCF and Landsat TCC also suggest that by training TCC with VCF tree cover these biases have been propagated into the finer-scale product. Models calibrated using MODIS VCF (Brandt et al., 2017; Lasslop et al., 2020; Burton et al., 2019; Kelley et al., 2019, 2021) also risk inheriting these biases and should therefore be validated using other sources of data. "*

We also highlight and incorporate in our introduction and discussions the only other Landsat TCC evaluation carried out by Montesano et al., (2016):

Introduction:

*"Likewise, validation of the finer-scale TCC product has been limited to its penultimate version and to the taiga-tundra circumpolar region (Montesano et al., 2016)."*

*"Similarly to MODIS VCF (Montesano et al., 2009), Montesano et al., (2016) revealed an overestimation of the taiga-tundra low tree covers in the finer-scale Landsat TCC, suggesting that using VCF as training has propagated these overestimations into the higher resolution product."*

Discussions:

*"According to its definition, MODIS VCF only maps trees that are 5 m or taller (Hansen et al. 2003), while the TROBIT CAI includes all trees with a minimum dbh of 2.5 cm, as well as trees with a height exceeding 1.5 m when dbh < 2.5 cm. This could explain our observed underestimation in the lower tree cover ranges for both MODIS VCF and Landsat TCC. In fact Montesano et al. (2016) showed an improved match between Landsat TCC and their lidar-derived tree cover reference data when reducing the height threshold from 5 m to 2 m. However, because of how our field*

*reference CAI is derived, we were not able to conclusively link the 5 m threshold to our observed underestimation."*

*"To ensure the appropriate use of both products, we suggest that where field data are available, the products should be calibrated for use in the target region. However, calibrating on a large scale using field data as a reference presents several challenges. Firstly, different in-situ measurement techniques tend to measure different types of tree cover (e.g. Fiala et al., 2006; Korhonen et al., 2006; Rautiainen et al., 2005) and each will require a specific conversion method to enable direct comparison with MODIS VCF or Landsat TCC. For example, Montesano et al. (2016) 's comparison did not acknowledge VCF's and thus TCC's 'within canopy gaps,' which may explain their observed underestimation in covers above 80%. "*

Reviewer2:
Thank you to reviewer #2 for your further review. In response to your concerns about using a limited number of sites to represent a very varied tropical savanna-forest ecotone, we have further edited the text to highlight the limitations of the global extrapolation as follows:

We have altered the text in the introduction to: *"In this study, we evaluate MODIS VCF Collection 6 in tropical savannas and forest areas by comparing VCF's tree cover percentage to corresponding field data."*
*"We then, for MODIS VCF, characterise the observed bias in woody covers across both savanna and forest ecosystems and apply our calibration across the tropics to highlight the regions most likely affected by these inaccuracies. We finish by discussing the implications the uncovered biases may have on tropical vegetation and terrestrial biogeochemical modelling."*

We also refer to 'calibration' instead of 'correction' throughout the text. We have also replaced Figure 2 with a new figure (Figure 5) that combines Figure 2 and Figure A7 and now shows (i) the post-calibration change in tree cover that is statistically significant across all four scenarios, (ii) the uncertainty range of the calibration, (iii) priority regions for field surveying. The accompanying text in the results and parts of the discussion further stress the uncertainty of our calibration.

---

## Author Response (AR3)

Dear Editor, please find attached our revised manuscript. We have dealt with all the reviewer's comments and your recommendations and hope this improved version has addressed all the outstanding issues.

As per your recommendation, we have included the map showing distance to the field plots in Figure 6.

Regarding the reviewer 1's comments:

1) Raise the importance of Figure A1 (Location of sampling sites in Africa, Australia, and South America from the TROBIT Project). I suggest moving the map in Figure A1 as a main figure (Figure 1) of the paper. This will also help readers to better understand the spatial patterns shown in Figure 5, especially the priority map at the bottom.

We have moved Figure A1 into section 2.1 of the Methods.

2) Add more explicit discussion on the uncertainties of the field data used in this study for VCF evaluation, in e.g. lines 477-479. When discussing the various uncertainty factors of the presented analysis, readers should be reminded of the limited number of field sites and their spatial distribution (I understand that this has already been briefly mentioned above in the Results section in lines 356-357), as well as the uncertainties associated with allometric equations for calculating CAI from DBH. The authors are also suggested to extend the discussion by sharing thoughts on the implications for future in situ data collection (e.g. directly measure tree canopy cover from the field).

We have amended the Discussion to include additional clarification on the associated uncertainties. The revised version is as follows:

[There is also the uncertainty associated with the field data collection. In our case, the site-specific CAI standard errors (supplement B in Torello Raventos et al., 2013) are small and show no systematic bias, and are therefore not expected to significantly change our results. However, our results in Figure 6  have been extrapolated from a limited number of field sites, with somewhat limited distribution across the tropics. While our uncertainty map gives a good idea of areas of concern, for a more robust description of MODIS VCF's accuracy, we would need substantially more widely distributed  in-situ sites to represent the tropics. Field data collection remains costly and labour intensive and adopting protocols that ensure standardisation and data sharing via information platforms would facilitate large-scale validation exercises. In the case of validating woody cover products, including direct tree cover measurements such as crown area would also help.

However, the reality for much of pre-existing data is that they are not standardised, and field data (whether standardised or not) are not designed to be directly comparable to remote sensing-derived variables. Here, our technique offers a potential solution to this. By accounting for uncertainties that arise from differences between the available in-situ data and the remotely sensed variable we are able to use in-situ data, which may initially seem unsuitable, to carry out a more conservative evaluation. ]

Lines 44-45 The sentence "We estimate that MODIS VCF could be underestimating tropical tree cover by as much as 29%" is an exaggeration, as it represents the very extreme case of the wide ranges and scenarios of the findings from this particular study (7-29% or 0-21%, depending on analysis scenarios, lines 394-397). Thus, the exaggerated statement should be removed from the abstract.

The lines describing our results in the abstract has been amended to

 [Our scenarios suggest that MODIS VCF accuracy can vary substantially, with tree cover underestimation ranging from 0 to 29 %. ]

Lines 456-458. "Models calibrated using MODIS VCF (Brandt et al., 2017; Lasslop etal., 2020; Burton et al., 2019; Kelley et al., 2019, 2021) also risk inheriting these biases and should therefore be validated using other sources of data." This generation is beyond the current analysis, and is more likely wrong than correct. Recommend deleting this statement from the manuscript.

We amended the line to:

[There is a risk of bias propagation when MODIS VCF (or related products) is used as a single source for benchmarking models (e.g.  Brandt et al., 2017; Lasslop et al., 2020; Burton et al., 2019; Kelley et al., 2019, 2021). To avoid this, studies should try use multiple data sources (e.g. for woody cover possibly the GEDI lidar based canopy cover product, Tang et al 2019) whether for model calibration/validation (e.g. Sellar et al., 2019 and Wiltshire et al., 2020, Burton et al., 2021)  or hypothesis testing  (e.g. Taylor et al., 2012).]

We are not clear what the reviewer means by ' more likely wrong than correct'. We believe it is important to highlight to the readers the risk associated with using a biased product as the sole source for benchmarking a model or testing a hypothesis. We hope the introduced changes have helped clarify our message.

However, if the removal of this line is essential to secure the acceptance of our paper, we are happy to oblige and take it out altogether.